# Analysis of Voltage Control Strategies for DC Microgrid with Multiple Types of Energy Storage Systems

Zhichun Yang [1,2], Chenxia Wang [3], Ji Han [3], Fan Yang [1,2], Yu Shen [1,2], Huaidong Min [1,2], Wei Hu [1,2] and Huihui Song [3,*]

1   Electric Power Research Institute, State Grid Hubei Electric Power Co., Ltd., Wuhan 430077, China
2   Key Laboratory of High-Voltage Field-Test Technique of SGCC, Wuhan 430077, China
3   College of New Energy, Harbin Institute of Technology at Weihai, Weihai 264200, China
*   Correspondence: songhh@hitwh.edu.cn

**Abstract:** Direct-current (DC) microgrids have gained worldwide attention in recent decades due to their high system efficiency and simple control. In a self-sufficient energy system, voltage control is an important key to dealing with upcoming challenges of renewable energy integration into DC microgrids, and thus energy storage systems (ESSs) are often employed to suppress the power fluctuation and ensure the voltage stability. In this paper, the performances of three voltage control strategies for DC microgrids are compared, including the proportion integration (PI) control, the fuzzy PI control and particle swarm optimization (PSO) PI control. Particularly, two kinds of ESSs including battery and advanced adiabatic compressed air energy storage (AA-CAES) with different operational characteristics are installed in the microgrid, and their impacts on voltage control are investigated. The control performances are comprehensively compared under different control schemes, various scenarios of renewable energy fluctuations, participation in the control of the two ESSs or not, and different fault conditions. Additionally, the dynamic performances of the ESSs are exhibited. The results verify the validity of the control schemes and the feasibility of the configuration of the ESSs into the DC microgrid.

**Keywords:** AA-CAES; DC microgrid; voltage stabilization control; droop control; fuzzy PI control; PSO PI control

## 1. Introduction

Microgrids can combine distribution networks, renewable energy power supply systems, energy storage systems, local loads, etc., and represent important parts of the energy Internet. The existing microgrid project mainly uses an alternating-current (AC) microgrid, which is compatible with the AC power grid and AC power equipment. However, the output of photovoltaic power generation, fuel cells and other renewable energy sources is DC. In order to adjust the output frequency, wind power generation, gas turbine and other renewable energy sources usually convert AC output into DC for further use [1]. The output of many energy storage batteries, such as batteries and lithium batteries in energy storage devices, is also DC. At the same time, DC electrical equipment such as electronic equipment, electric vehicles, LED lighting equipment are increasingly widely used. Therefore, the above DC source load storage is directly connected to the DC converter through the DC bus to form a DC microgrid, which can effectively reduce the cost and power loss of intermediate conversion linking, and improve the power supply quality [2]. Compared with AC microgrids, DC microgrids have many advantages, such as high efficiency, high reliability, high controllability, low cost, etc. It is an ideal scheme for the further development of microgrid.

Compared with AC microgrids, DC microgrids have no problems in reactive power, phase and frequency, and DC voltage has become an important indicator of system stability [3]. In DC microgrid system, in order to ensure the stability of DC voltage, it is

necessary to coordinate the control of multiple microsources. Reference [4] proposed a voltage-layered coordination control strategy. The control strategy coordinates the working mode of power electronic converters by detecting the variation in DC voltage and maintains the stability of DC voltage. However, under the same working conditions, only one inverter in the system uses the droop characteristic to participate in voltage regulation. Due to the limitation of the inverter's own capacity, the voltage regulation capability of the system is low. In order to improve the overall voltage regulation capability of the system, multiple inverters are required to participate in voltage regulation at the same time. Reference [5] proposed an improved droop control method based on the residual capacity of an energy storage unit. The droop coefficient is set according to the residual capacity of the energy storage unit to realize the coordinated control of DC voltage between distributed energy storage units. Due to the diversity of the main voltage-regulating units in the microgrid, it is necessary to adjust the DC voltage with multiple different main voltage-regulating units and at the same time to improve the overall voltage-regulating capability of the system. Each main voltage-regulating unit can maintain the power balance of the system according to its droop characteristics by setting different droop coefficients, so as to achieve the purpose of coordinated control of DC voltage at the same time. Reference [6] analyzed the stability of DC microgrids after droop control through eigenvalue loci and used impedance matching criteria to increase active damping through low-pass filtering to suppress voltage oscillation. Reference [7] considered power filtering, droop characteristics and other factors, and analyzed the influence of constant power load and droop coefficient on system stability through linearization. References [8,9] established a small-signal model based on the characteristic curve of constant power load, and obtained the conditions that the system needs to meet to achieve stable operation by analyzing the voltage transfer function of load and power side. A multi-source coordinated control strategy based on voltage slopes is proposed in reference [10]. This strategy includes using the voltage-stabilizing control strategy of section I at the moment of voltage drop at the grid side, the voltage-stabilizing control strategy of section II after the energy storage is out of operation and the corresponding secondary control. The strategy proposed in this paper can effectively accelerate the bus stability time and reduce its fluctuation, but the selection of energy storage unit is still based on the battery energy storage. Reference [11] proposed the use of three evolutionary search algorithms, particle swarm optimization (PSO), simulated annealing (SA) and genetic algorithms (GA), to determine the optimal parameter values of the fractional-order (FO)-PI controllers implemented in dual-active bridge-based (DAB) DC microgrids. The aim of this was to find the optimal parameters for these controllers in terms of voltage stability. Reference [12] proposed a passivity-based nonlinear control for an isolated microgrid system. The microgrid consists of a photovoltaic array and battery energy storage connected to a point of common converters, supplying a constant power load. The purpose of this control strategy is to maintain the output direct current voltage in its reference value under load variations, improving battery interaction. Reference [13] proposes a DC-side synchronous active power control for two-stage photovoltaic power generation without energy storage, which can keep the DC link voltage stable under the condition of changing light intensity.

Distributed energy is booming, and these distributed power generation units are widely integrated into microgrids. Renewable energy is usually highly dependent on weather conditions, resulting in a certain degree of intermittence and uncertainty, which brings pressure to the operation and control of microgrids. Reference [14] realized extremely low-frequency energy harvesting by utilizing an escapement mechanism to achieve frequency tuning. Reference [15] proposed an optimal planning method considering wind power and photovoltaic power limiting measures to optimize the optimal planning of DG integration location and optimal capacity, and to formulate consumption plans for each period.

Renewable energy generally exists in the microgrid, and energy storage is usually required to decouple fluctuating energy supply and rigid energy demand, improve the flex-

ibility of the system, and ensure the reliable operation of the system. As an indispensable part of the microgrid, the energy storage system is of great significance for improving the power quality, stability and operating efficiency of the microgrid, as well as achieving peak shaving and valley filling. However, under the current technical conditions, single-energy storage methods, such as existing batteries, supercapacitors, flywheel energy storage, superconducting energy storage, etc., which are suitable for microgrid applications, cannot simultaneously take into account the needs of energy density, power density and energy storage time. As a result, composite energy storage technology has come into being [16]. Reference [17] considered the battery degradation and thermal runaway propagation, established a multi-state model of BESS, and studied the reliability of large-scale grid-connected BESS and its impact on the overall reliability of power system. Reference [18] proposed a power distribution method of a composite energy storage system based on the state of charge by combining battery and super capacitor. Reference [19] used second-order filtering to separate wind power, and distributed high-frequency power to supercapacitors and intermediate-frequency components to batteries for stabilization. However, this method had phase lag characteristics when extracting grid connected power, which increased energy storage output. Reference [20] used a wavelet transform to decompose wind power, and determined the boundary frequency of energy type and power type compensation. Low-frequency power components are taken as grid connected power, while medium-frequency and high-frequency power components are compensated by energy type and power type energy storage. On this basis, reference [21] conducts power allocation based on empirical mode decomposition, and allocates energy type and power type energy storage compensation power based on minimum mode aliasing. However, the above research only studied the combination of energy storage and power storage.

According to the duration requirements of energy storage application scenarios, they can be divided into capacity type, energy type, power type and standby type. Among them, the capacity type energy storage belongs to long-term energy storage technology, which generally requires the energy storage duration to be no less than 4 h. With the increase in the proportion of new energy, such as wind and light, in the power generation structure, in order to ensure the stability of power supply, the power system has increasingly high requirements for the energy storage duration, and the demand for capacity type energy storage is growing. Compressed air energy storage uses green, rich and convenient air as the medium with which to skillfully solve the problem of time and space contradiction in the utilization of electric energy. At the same time, it can also combine the intermittent power generated by renewable energy to improve the quality of electric energy. It has the advantages of large energy storage capacity, long energy storage period and small specific investment. It is considered to be one of the most promising large-scale energy storage technologies; its application and development have attracted the attention of the international community [22]. In particular, the AA-CAES system, which has been vigorously developed in recent years, has overcome the dependence of early compressed air energy storage power stations on fossil fuels and completely achieved zero emissions. At the same time, the use of high-pressure gas tank gas storage technology [22,23] not only has little special requirements for environmental site selection, but also greatly reduces the overall volume of the energy storage device, which can achieve miniaturization, thus providing ideal capacity type energy storage equipment for the microgrid. References [24,25] discussed the role of compressed air energy storage (CAES) system in coordinated dispatching, phase modulation and other auxiliary services. Reference [26] studied the unit commitment of power systems with a CAES power station and wind power grid connection, and made specific analysis from many aspects such as wind power consumption, congestion management, peak shaving and valley filling. References [27] and [28], respectively, put forward a capacity optimization configuration method for isolated microgrids with a CAES power station and wind power generation system with an AA-CAES power station. With the goal of minimizing the sum of system operation cost and environmental cost, as well as the sum of user interruptible load cost and system comprehensive cost, the optimal

configuration capacity of a CAES/AA-CAES power station, a wind turbine generator unit and other units was obtained. Reference [29] addressed the development of the energy compensation method used for the design a hybrid energy storage system (HBESS). The study examined the energy density that each chemical element carries in its structure, as well as the electrochemical cell that can be composed of one or more elements. The specific energy difference between two battery technologies makes it possible to determine which compensation factor should be adopted for each technology. This method was developed to meet the demand for substations outside the Brazilian standard of power systems, which lack an uninterrupted and reliable energy source. The method was validated by designing a microgrid to support the auxiliary systems of a transmission substation in northeastern Brazil. The operation and control strategy of energy storage systems, intended for application in hybrid microgrids with AC coupling, was proposed in reference [30]. By adding an energy storage system, this strategy, based on optimized indirect control of diesel generators, seeks to increase generation efficiency, reduce working time, and increase the introduction of renewable sources in the system. As a result, there is a significant reduction in diesel consumption, a decrease in the power output variance of the diesel generation system, and an increase in the average operating power, which ensures effective control of hybrid plants. In reference [31], battery energy storage systems (BESSs) were used to provide energy to users in cases of power failure or major energy quality problems, and the test results of different load configurations of microgrids at the time of interruption were given. The functionality of the BESS was evaluated in the face of real outages that occurred in a microgrid located in the Brazilian countryside. The results presented demonstrated the effectiveness of BESS in the energy supply of microgrid loads. Reference [32] studied the power quality issues related to wind turbines and power systems, as well as how (BESS) can solve or mitigate these disturbances in the network. The research into wind farms provided the results of five applications (factor correction, voltage control, power factor smoothing, frequency control and time shift). A BESS was paralleled with a wind turbine. The effectiveness of the BESS was verified by the actual operation data of each application.

To sum up, for composite energy storage, most of the existing studies only study power type and energy type composite energy storage systems, without considering capacity type energy storage units. For AA-CAES systems, most existing studies connect AA-CAES systems to AC microgrids, and few articles consider connecting an AA-CAES system to DC microgrids to participate in the DC microgrid voltage regulation process. This paper focuses on the operation control strategy for AA-CAES and the participation of battery composite energy storage p in DC microgrid voltage stabilization.

In this paper, the DC microgrid voltage stabilization control is studied, and a composite energy storage scheme consisting of AA-CAES and battery technology is proposed. A DC microgrid voltage stabilization control strategy is designed based on droop control and improved PI control, which effectively improves the stability of DC microgrid operation. The simulation model of a DC microgrid system with composite energy storage is built on a simulation platform. The proposed control strategy can help to improve the voltage stability under the circumstances of light intensity fluctuation and power generation unit failure.

## 2. DC Microgrid System Structure and Dynamic Modeling of Its Constituent Units

### 2.1. DC Microgrid System Structure

Figure 1 shows the structural diagram of the DC microgrid system with AA-CAES and battery energy storage. The diesel generator and AA-CAES system are connected to the DC bus of the microgrid through an AC/DC converter, and the battery is connected to the DC bus through an DC/DC converter. The DC microgrid in this paper adopts a single-bus structure, and all power generation units, energy storage units and loads are connected to the same DC bus. Under this configuration, the location and placement of the ESS may have little influence on its ability to participate in grid operations, and so the optimum location and placement of the ESS has not been considered. In the figure, $P_G$, $P_C$,

$P_{PV}$ and $P_B$ are the output power of the diesel generator set, AA-CAES, photovoltaic power generation unit and battery, respectively.

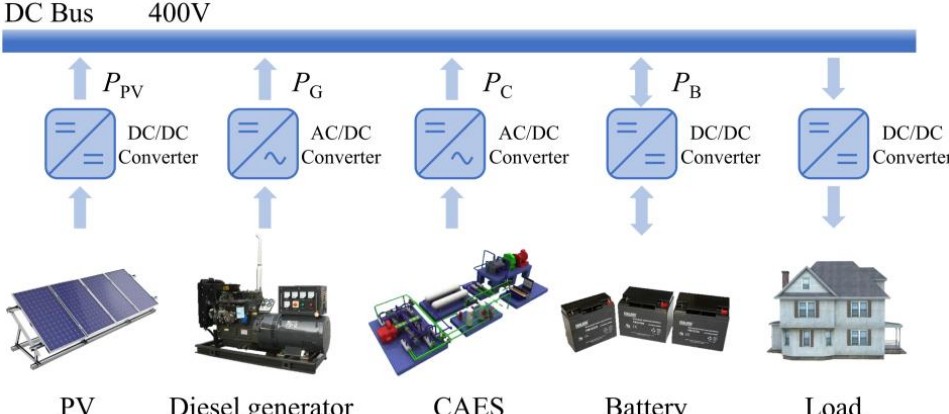

**Figure 1.** DC microgrid system structure.

A fixed-line rating system is adopted in this paper, and an elastic rating system can be added in subsequent research to improve the control performances. For example, instead of limiting the line rating to the fixed value in static line rating (SLR) based on worst conditions, future studies may use the line monitoring sensor and weather monitoring system to provide the actual line operating condition and improve line performance. More studies of elastic rating system can be found which: ① Consider line natural aging and line load line fault based on a dynamic thermal rating system [33]. ② Combine the dynamic thermal rating system with BESS to carry out line rating and stabilize the output of renewable energy to improve the reliability of power grid [34]. ③ Evaluate the reliability of the model when using combined with network topology optimization technology, a dynamic thermal rating system and a battery storage system [35].

### 2.2. Dynamic Modeling and Characteristic Analysis of Each Unit in DC Microgrid

2.2.1. Modeling of Photovoltaic Power Generation Unit

The equivalent circuits of the solar photovoltaic system and DC/DC boost converter are shown in Figure 2 [36]. In the figure, $V_{PV}$ is the output voltage of the solar photovoltaic unit, $I_{PV}$ is the output current of the solar photovoltaic unit, $C_{PV}$ is the filter capacitor, $L_{PV}$ is the inductance, $R_{PV}$ is the internal resistance, $I_L$ is the inductance current, $C_{dc}$ is the DC bus capacitance, $V_{dc}$ is the DC bus voltage, $I_{dc}$ is the output current of the converter, and $\mu$ is the duty cycle of the converter MOSFET switch.

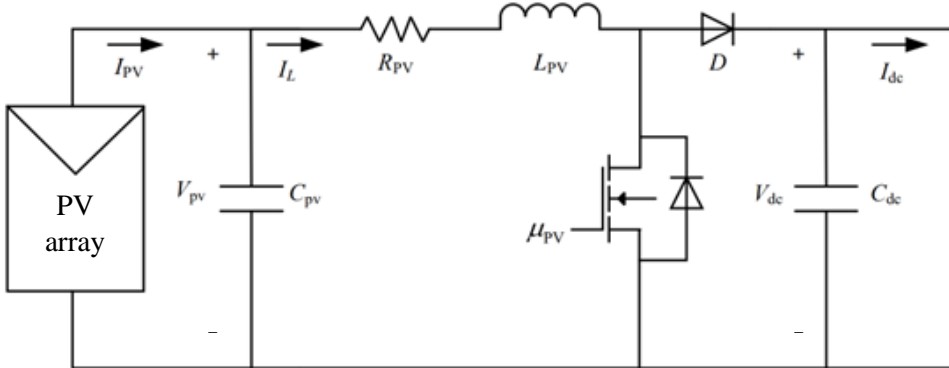

**Figure 2.** Equivalent circuit diagram of solar PV system and DC/DC boost converter.

According to Figure 2, the dynamic model of the solar photovoltaic system connected to the DC/DC boost converter can be described with the following differential equation:

$$\dot{V}_{PV} = \frac{1}{C_{PV}}(I_{PV} - I_L) \tag{1}$$

$$\dot{I}_L = \frac{1}{L_{PV}}[-R_{PV}I_L + V_{PV} - (1 - \mu_{PV})V_{dc}] \tag{2}$$

$$\dot{V}_{dc} = \frac{1}{C_{dc}}(1 - \mu_{PV})I_L - \frac{1}{C_{dc}}I_{dc} \tag{3}$$

### 2.2.2. Modeling of Diesel Generator

A diesel generator is a synchronous generator with an exciter and governor which is driven by a diesel engine. The excitation system of the synchronous generator is used to maintain the desired voltage at the generator terminals. In the same way as above, the two-axis model of synchronous generator and the dynamics of excitation system can be expressed with the following formula [37]:

$$\dot{\delta} = \omega - \omega_0 \tag{4}$$

$$\dot{\omega} = -\frac{D}{2H}(\omega - \omega_0) + \frac{\omega_0}{2H}P_m - \frac{\omega_0}{2H}\left(E'_q I_q + E'_d I_d\right) \tag{5}$$

$$\dot{E}'_q = -\frac{1}{T'_{do}}E'_q - \frac{(x_d - x'_d)}{T'_{do}}I_d + \frac{1}{T'_{do}}E_{fd} \tag{6}$$

$$\dot{E}'_d = -\frac{1}{T'_{qo}}E'_d + \frac{(x_q - x'_q)}{T'_{qo}}I_q \tag{7}$$

$$\dot{E}_{fd} = -\frac{E_{fd}}{T_A} + \frac{K_A}{T_A}(V_{ref} + V_C - V_t) \tag{8}$$

where $\delta$ is the rotor angle, $\omega$ is the working speed of the engine, $\omega_0$ is the synchronous speed, $D$ is the damping coefficient, $H$ is the inertia constant, $P_m$ is the mechanical power input, $E'_q$ is the magnetic field variable proportional to the flux linkage, $T'_{do}$ is the *d*-axis transient open-circuit time constant, $x_d$ is the *d*-axis synchronous reactance, $x'_d$ is the d-axis transient reactance, $I_d$ is the *d*-axis current component, $E_{fd}$ is the equivalent excitation voltage, $E'_d$ is the damper variable proportional to the *d*-axis damper flux linkage, $T'_{qo}$ is the *q*-axis transient open-circuit time constant, $x_q$ is the *q*-axis synchronous reactance, $x'_q$ is the *q*-axis transient reactance, $I_q$ is the *q*-axis current component, $T_A$ is the time constant of the voltage regulator, $K_A$ is the gain of the voltage regulator, $V_{ref}$ is the reference terminal voltage, and $V_C$ is the stable signal and control input of the system.

The terminal voltage of generator is shown in Formula (9):

$$V_t = \sqrt{\left(E'_q - x'_d I_d\right)^2 + (x'_d I_q)^2} \tag{9}$$

### 2.2.3. Modeling of Battery Energy Storage

The equivalent circuit diagram of the energy storage system with a bidirectional DC/DC converter is shown in Figure 3, and its dynamic model is established [36].

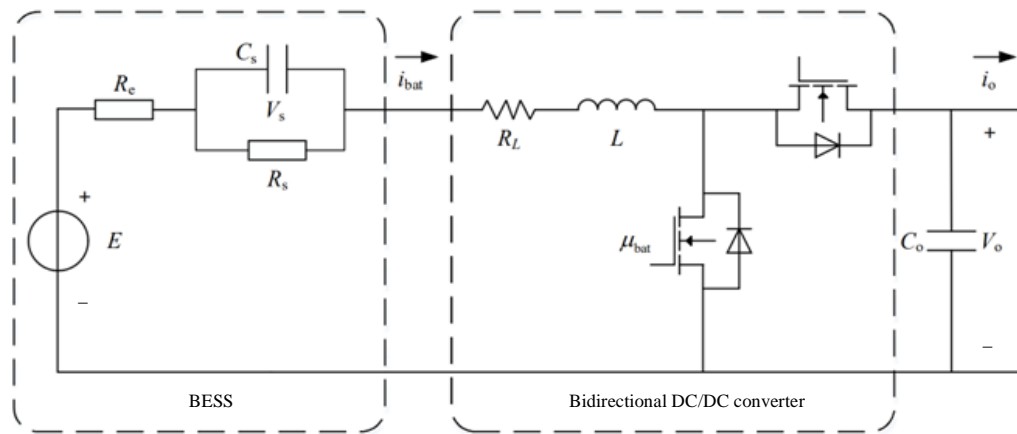

**Figure 3.** Equivalent circuit diagram of battery energy storage system and bidirectional DC/DC buck–boost converter.

Based on Figure 3, the dynamics of the energy storage system and bidirectional converter can be described with the following nonlinear state space average differential equation:

$$\frac{dV_s}{dt} = \frac{1}{C_s}\left(i_{bat} - \frac{V_s}{R_s}\right) \tag{10}$$

$$\frac{di_{bat}}{dt} = \frac{1}{L}\left[-(R_L + R_c)i_{bat} + (1 - \mu_{bat})V_o - V_s + E\right] \tag{11}$$

$$\frac{dV_o}{dt} = \frac{1}{C_o}\left[(1 - \mu_{bat})i_{bat} - i_o\right] \tag{12}$$

2.2.4. AA-CAES System Modeling

CAES technology is a technology that uses compressed air to store energy. Its working principle is that, during the period of low power consumption, the air is compressed to high pressure by electric energy and stored in the pressure vessel, so that electric energy can be converted into air energy for storage. During the peak period of power consumption, the high-pressure air is released from the gas storage chamber, enters the combustion chamber for combustion, heats up by fuel combustion, and then drives the turbine to generate electricity. The AA-CAES technology combines heat storage technology with the traditional CAES technology. Compared with the traditional supplemental combustion energy storage system, the heat storage device replaces the combustion chamber and uses the heat storage device to recover and reuse the compressed heat. The heat storage device stores the heat generated in the process of compressed air, releases the stored heat in the turbine energy release stage, and returns it to the compressed air in order to increase the output work of the turbine, get rid of the dependence on fossil fuels, and improve the overall efficiency of the system. The main equipment of AA-CAES technology includes a compressor, expander, gas chamber, heat accumulator and heat exchanger. The AA-CAES system is shown in Figure 4.

1.    Piston compressor model

The dynamic modeling process of piston compressor is based on the following two assumptions: (1) air is an ideal gas; (2) there is no heat exchange between the cylinder and the external environment. The geometric structure of the piston compressor is shown in Figure 5. $L_{com}$ and $R_{com}$ are, respectively, the length of the connecting rod and the radius of the crank.

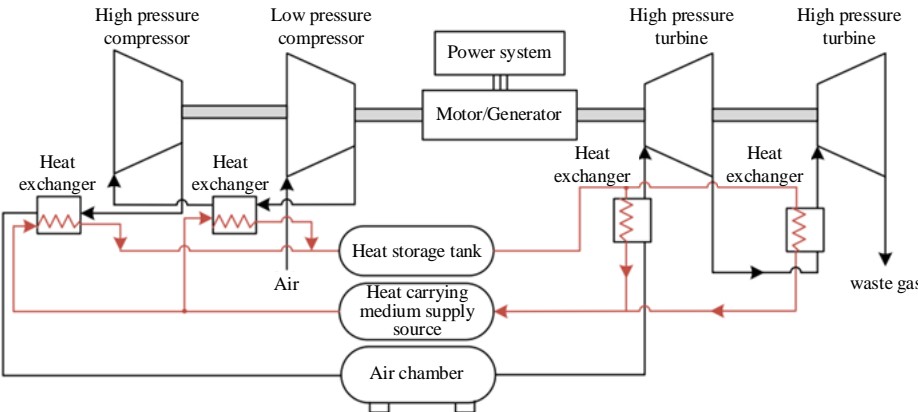

**Figure 4.** AA-CAES system structure diagram.

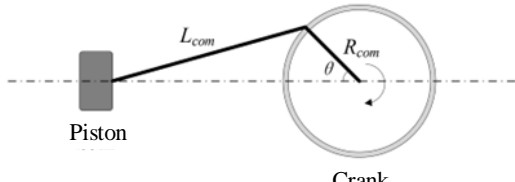

**Figure 5.** Schematic diagram of geometric structure of piston compressor.

According to the first law of thermodynamics, it can be obtained that the change rate $\dot{p}_{cy}$ of air pressure in the cylinder is [38]:

$$\dot{p}_{cy} = \frac{\gamma}{v_{cy}} \left( R_g T_{com,in} \dot{m}_{in,com} - R_g T_{cy} \dot{m}_{out,,com} - p_{cy} \dot{v}_{cy} \right) \tag{13}$$

where, $\gamma$ is the specific heat ratio of air; $v_{cy}$ and $\dot{v}_{cy}$ are the change rate of cylinder volume and cylinder volume, respectively; $T_{com,in}$ and $T_{cy}$ are the compressor inlet gas temperature and the air temperature in the cylinder, respectively. $T_{cy}$ can be calculated from the ideal gas state equation; $\dot{m}_{in,com}$ and $\dot{m}_{out,,com}$ are the intake air mass flow rate and the storage air mass flow rate of the cylinder, respectively. Their values can be calculated with the orifice throttling theory [39].

2. Radial turbine model

The torque acting on the radial turbine rotor mainly includes two items: (1) unit mass flow torque, caused by sudden the deflection of air flow when entering the rotor; (2) unit mass flow torque, generated by air flow in the rotor flow passage.

The air mass flow $\dot{m}_N$ through the turbine stator and the air mass flow $\dot{m}_R$ through the rotor are, respectively, expressed as follows [40]:

$$\dot{m}_N = c_{tur,2} \sin \alpha_2 \pi d_{tur,2} b_{tur,2} \rho_{tur,1} \left( \frac{p_{tur,2}}{p_{tur,2}} \right)^{\frac{1}{\gamma}} \tag{14}$$

$$\dot{m}_R = c_{tur,3} \sin \alpha_3 \pi d_{tur,3} b_{tur,3} \rho_{tur,2} \frac{T_{tur,2}}{T'_{tur,2}} \left( \frac{p_{tur,3}}{p_{tur,2}} \right)^{\frac{1}{\gamma}} \tag{15}$$

where, $c_{tur,2}$ and $c_{tur,3}$ represent the absolute speed at the rotor inlet and the absolute speed at the rotor outlet, respectively; $\alpha_2$ is the absolute air flow angle at the rotor inlet; $d_{tur,2}$ and $d_{tur,3}$ represent the outer diameter and average inner diameter of the rotor, respectively; $b_{tur,2}$ and $b_{tur,3}$ represent the blade width at the rotor inlet and the blade height at the rotor outlet, respectively; $p_{tur,1}$ and $p_{tur,2}$ represent the air pressure at the stator and rotor inlet, respectively; $p_{tur,3}$ indicates the air pressure at the turbine outlet; $\rho_{tur,1}$ and $\rho_{tur,2}$ represent the average gas density in the stator and rotor flow channels, respectively; and $T_{tur,2}$ and

$T'_{\text{tur},2}$ represent the rotor inlet temperature and the rotor inlet temperature that changes due to the impact angle loss, respectively.

According to the mass conservation theorem, the gas mass flow through the stator and rotor should be equal. Therefore, the problem of solving the nonlinear equations is to find a rotor inlet air pressure $p_{\text{tur},2}$, that makes $\dot{m}_{\text{N}} = \dot{m}_{\text{R}}$.

3.　Air chamber model

The heat transfer process between the gas storage chamber and the external environment is considered in the gas storage chamber model, and the mass loss in the gas storage chamber is not considered.

The change rate of air pressure in the air chamber is shown as follows [41]:

$$\dot{p}_{\text{st}} = \frac{R_{\text{g}}\gamma T_{\text{in}}}{V_{\text{st}}}\dot{m}_{\text{in,st}} - \frac{R_{\text{g}}\gamma T_{\text{st}}}{V_{\text{st}}}\dot{m}_{\text{out,st}} - \varsigma_{\text{eff}}(T_{\text{st}} - T_{\text{wall}}) \tag{16}$$

where, $\dot{m}_{\text{in,st}}$ and $\dot{m}_{\text{out,st}}$ represent the gas mass flow into and out of the gas chamber, respectively; $V_{\text{st}}$ is the volume of the air chamber; $T_{\text{st}}$ and $T_{\text{wall}}$ represent the air temperature in the air chamber and the temperature of the air chamber wall, respectively; $\dot{p}_{\text{st}}$ is the change rate of air pressure in the air chamber; and $\varsigma_{\text{eff}}$ represents the equivalent heat transfer coefficient between the air chamber and the outside world.

4.　Shell and tube heat exchanger model

Shell and tube heat exchanger includes shell side and tube side. The fluid in the shell side is recorded as fluid 1, and the fluid in the tube side is recorded as fluid 2. The temperature dynamic equation of shell-side fluid is shown in Formula (17) [41].

$$\frac{\partial T_{\text{HEX},1}}{\partial t} = -\frac{\dot{m}_{\text{HEX},1}}{m_{\text{HEX},1}}\frac{\partial T_{\text{HEX},1}}{\partial x} + \frac{K_{\text{o}}n_{\text{HEX}}\pi d_{\text{o}}}{m_{\text{HEX},1}c_{\text{p,fluid1}}}(T_{\text{HEX,wo}} - T_{\text{HEX},1}) \tag{17}$$

where, $T_{\text{HEX},1}$ is the temperature of shell-side fluid; $m_{\text{HEX},1}$ is the mass per unit length of fluid on the shell side; $\dot{m}_{\text{HEX},1}$ is the fluid mass flow through the shell side; $c_{\text{p,fluid1}}$ are the specific heat capacity of the shell-side fluid; $T_{\text{HEX,wo}}$ is the temperature of the outer wall of the pipe; $d_{\text{o}}$ is the outer diameter of the heat exchange tube; $n_{\text{HEX}}$ is the number of heat exchange tubes; and $K_{\text{o}}$ is the shell-side heat transfer coefficient.

Temperature dynamic equation of pipe-side fluid:

$$\frac{\partial T_{\text{HEX},2}}{\partial t} = \frac{\dot{m}_{\text{HEX},2}}{m_{\text{HEX},2}}\frac{\partial T_{\text{HEX},2}}{\partial x} - \frac{K_{\text{i}}n_{\text{HEX}}\pi d_{\text{i}}}{m_{\text{HEX},2}c_{\text{p,fluid2}}}(T_{\text{HEX},2} - T_{\text{HEX,wi}}) \tag{18}$$

where, $T_{\text{HEX},2}$ is the temperature of the fluid in the tube side; $T_{\text{HEX,wi}}$ is the temperature of the inner wall of the pipe; $m_{\text{HEX},2}$ is the mass per unit length of pipe-side fluid; $\dot{m}_{\text{HEX},2}$ is the fluid mass flow through the tube side; $K_{\text{i}}$ is the heat transfer coefficient of the tube side; $c_{\text{p,fluid2}}$ is the specific heat capacity of the fluid in the tube side; and $d_{\text{i}}$ is the inner diameter of the heat exchange tube.

## 3. Control Strategy of Independent DC Microgrid with Composite Energy Storage

### 3.1. Control Strategy of DC Microgrid System

This section mainly introduces the control strategies adopted for each unit in the DC microgrid, including the photovoltaic power generation unit, diesel generator, battery, AA-CAES, and the online tuning and optimization method for PI parameters. The photovoltaic power generation system uses MPPT control, and the diesel generator and two kinds of energy storage units use traditional droop control methods. The control block diagram of the DC microgrid system is shown in Figure 6. Where $U_{\text{dc}}$ is the DC bus voltage and $U_{\text{ref}}$ is the DC bus voltage rating.

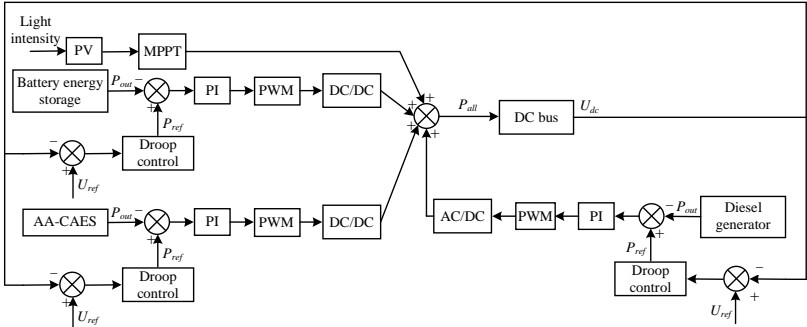

**Figure 6.** Control block diagram of DC microgrid system.

### 3.1.1. MPPT Control of Photovoltaic Power Generation Unit

This paper adopts the MPPT control strategy for photovoltaic power generation units. Under different external conditions, the P-V curves of photovoltaic cells are convex functions of single-peak values, and there are different and unique maximum power points. Therefore, in order to maximize the utilization of light energy, it is necessary to use the control system to adjust the operating state of photovoltaic cells to keep them at the optimal operating point. The corresponding control method is called the maximum power tracking control method, namely the MPPT control method.

This paper adopts the disturbance observation method for MPPT control, which is a widely used self-optimizing MPPT method. The existing literature has been introduced in detail, so this paper will not repeat this here.

### 3.1.2. Droop Control of Diesel Generator and Energy Storage Unit

In this paper, the droop control method is adopted for diesel generator and two kinds of energy storage units. The droop control method has two advantages. One is that there is no centralized controller or master–slave mechanism, and that all micro power sources are equal to each other in their capacity to jointly maintain the bus voltage and frequency, something which improves the anti-interference ability and reliability of the system. Second is that all micro power supplies work in droop mode, which can realize quick exit and access of micro power supplies, and improve the redundancy and scalability of the system.

In the DC microgrid, the traditional system droop control operation curve is shown in Figure 7, and mainly includes the droop charging and discharging mode and constant power charging and discharging mode. When the output power of the generating unit is lower than the constant power limit, the generating unit is in droop mode, and the output power is adjusted with the bus voltage change to maintain the bus voltage stability. When the output power exceeds the constant power limit, the power generation unit is in the constant power mode, and the output power is clamped at the maximum power for protection. The expression of traditional droop control is as follows:

$$P_{\text{ref}} = (v^* - v_{\text{dc}})\frac{1}{r_{\text{b}}} \tag{19}$$

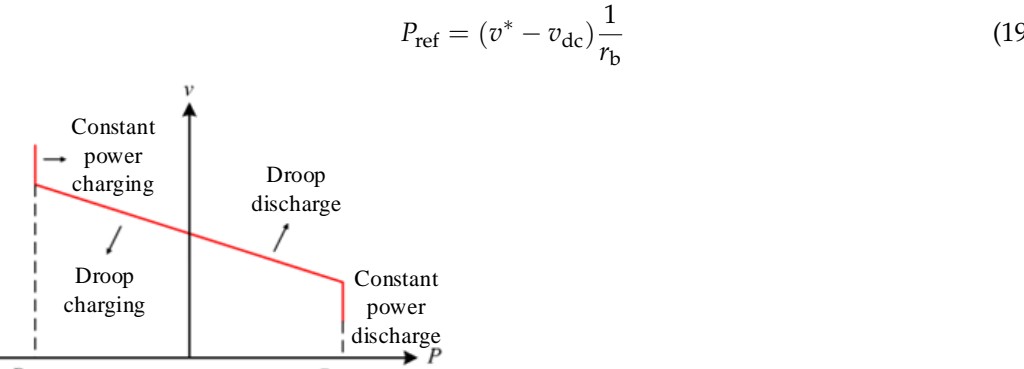

**Figure 7.** Droop control operation curve.

Droop control is described in detail in the existing data and this information will not be repeated in this paper.

### 3.2. PI parameter Tuning of System Controller

At present, the most commonly used control principles in the classical control theory used by the system are proportional, integral and PI control. PI control has the characteristics of simple principle, few controlled objects, convenient mathematical model establishment and easy analog or digital realization [42]. Therefore, at this stage, most of the control algorithms applied to the controlled objects are controlled by PI. However, although the PI regulator has good steady-state accuracy, it can only show such advantages in linearly controlled objects. In real application scenarios, most of the controlled objects are nonlinear systems due to external load interference. In the establishment of PI models, linear models that roughly discard nonlinear factors are often inaccurate, and dynamic performance and accuracy cannot be considered at the same time as the two are often contradictory. In view of the shortcomings of a PI controller, it is required that, in the process of PI control, the tuning of its parameters does not depend on the mathematical model and that the PI parameters can be adjusted online to meet the real-time control requirements. In this paper, fuzzy PI and PSO PI control methods are used to adjust PI parameters online.

#### 3.2.1. Fuzzy PI Control

Fuzzy PI control can realize the fuzzy self-adaptation of automatically adjusting PI control parameters, fuzzify accurate problems, and use fuzzy reasoning and defuzzification to achieve the optimal adjustment of PI parameters. Fuzzy adaptive PI control has many structure modes, but its principle remains fundamentally the same. This structure principle is shown in Figure 8.

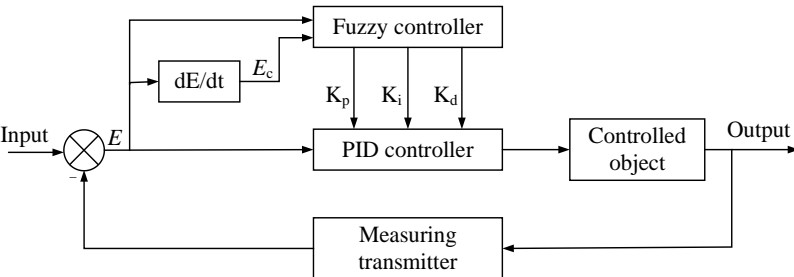

**Figure 8.** Structure of fuzzy adaptive PI controller.

Fuzzy adaptive PI control is composed of traditional PI control and fuzzy inference control. With deviation and deviation change rate as the inputs of a fuzzy controller, PI parameters are adaptively adjusted according to fuzzy control rules in order to meet the requirements of different deviation and deviation change rate on control parameters. As such, the organic combination of fuzzy control and PI control is truly realized [43].

For the DC microgrid system studied in this paper, $r(t)$ is the actual output power value of the generating unit, and $y(t)$ is the expected output power value after the application of droop control to the difference between the bus voltage and the rated value. Parameter $e(t)$ is defined as the difference between the actual output power and the expected output power, and $e_c(t)$ is the change rate of the difference between the actual output power and the expected output power, and a two-dimensional fuzzy controller is constructed as shown in Figure 9.

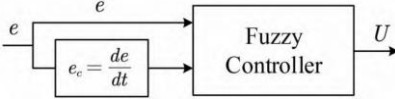

**Figure 9.** Structure of two-dimensional fuzzy controller.

The fuzzy controller takes the error $e$(t) and error transformation $e_c$(t) as the inputs, carries out fuzzy reasoning through the fuzzy controller, queries the fuzzy rule table for the defuzzifier and parameter adjustment, and then adjusts the PID parameters in real time according to the inputs of the error $e$(t) and error transformation $e_c$(t). According to the actual situation of the system studied in this paper:

The domain of the exact variable $e$(t) is: $[-3, 3]$;

The domain of the exact variable $e_c$(t) is: $[-3, 3]$;

Fuzzy sets of $e$(t): {NB, NM, NS, ZE, PS, PM, PB};

Fuzzy sets of $e_c$(t): {NB, NM, NS, ZE, PS, PM, PB}.

According to the above definition, we established the membership function of input and output variables in the fuzzy logic editor and formulated the fuzzy control rules. The fuzzy control rules are shown in Tables 1 and 2, and the fuzzy PI controller is designed.

**Table 1.** $\Delta K_\text{p}$ fuzzy rule table.

| $\Delta K_\text{p}$ | | de/dt Membership | | | | | | |
|---|---|---|---|---|---|---|---|---|
| | | **NB** | **NM** | **NS** | **ZE** | **PS** | **PM** | **PB** |
| | NB | PB | PB | PM | PM | PS | ZE | ZE |
| | NM | PB | PB | PM | PS | PS | ZE | NS |
| e | NS | PM | PM | PM | PS | ZE | NS | NS |
| membership | ZE | PM | PM | PS | ZE | NS | NM | NM |
| | PS | PS | PS | ZE | NS | NS | NM | NM |
| | PM | PS | ZE | NS | NM | NM | NM | NB |
| | PB | ZE | ZE | NM | NM | NM | NB | NB |

**Table 2.** $\Delta K_\text{i}$ fuzzy rule table.

| $\Delta K_\text{i}$ | | de/dt Membership | | | | | | |
|---|---|---|---|---|---|---|---|---|
| | | **NB** | **NM** | **NS** | **ZE** | **PS** | **PM** | **PB** |
| | NB | NB | NB | NM | NM | NS | ZE | ZE |
| | NM | NB | NB | NM | NS | NS | ZE | ZE |
| e | NS | NB | NM | NS | NS | ZE | PS | PS |
| membership | ZE | NM | NM | NS | ZE | PS | PM | PM |
| | PS | NM | NS | ZE | PS | PS | PM | PB |
| | PM | ZE | ZE | PS | PS | PM | PB | PB |
| | PB | ZE | ZE | PS | PM | PM | PB | PB |

The fuzzy controller is designed according to the rule table above. If it is detected that the given value of the output power is significantly different from the actual value (such as the starting moment), the power difference variable $e$(t) becomes the larger $\Delta K_\text{p}$ fuzzy rule table, and NB should be selected for e membership degree and NB for de/dt membership degree, after which the value of $\Delta K_\text{p}$ will be PB. That is, researchers should increase the value of the proportional coefficient $K_\text{p}$, so that the generating unit can respond quickly, increase the output power, and eliminate the error as soon as possible. If the difference between the given value of output power and the actual value is not large (such as stable operation and no external interference), then the power difference variable $e$(t) will be 0, according to $\Delta K_\text{p}$ fuzzy rule table, ZE will be selected for e membership, and ZE will be selected for de/dt membership, the value of $\Delta K_\text{p}$ will be ZE, and all power generation units will continue to operate stably in the current state.

### 3.2.2. PSO PI Control

PSO is an evolutionary algorithm, which originates from the imitation of the feeding behavior of birds. The idea of this algorithm is to treat the solution of the problem as being optimized as particles, where each particle corresponds to a fitness for evaluating the quality of the resulting particles. All particles together form a particle swarm. In order to

make particles move in the optimal direction, the concept of velocity is introduced. That is, velocity represents the direction and distance of movement, reflecting how far from the optimal solution. The moving speed is updated by tracking the individual extreme value Pt and the group extreme value Gt, moving in the solution space, and exiting the algorithm when the termination condition is met [44]. PSO has the advantages of easy realization and high efficiency, and so this paper uses PSO to optimize PI parameters. The principal process of using PSO algorithm to optimize PI controller is shown in Figure 10.

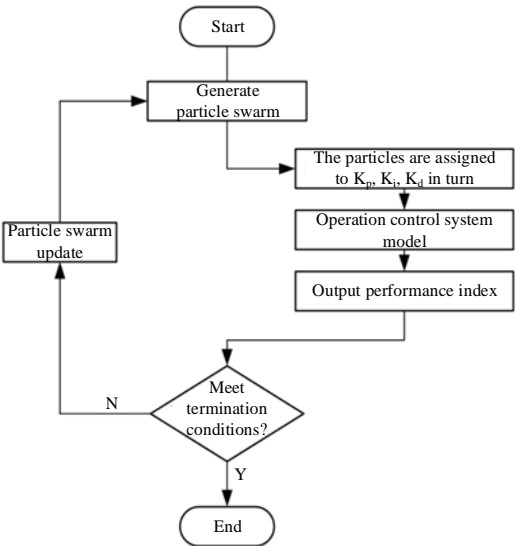

**Figure 10.** Schematic diagram of PSO PI principal process.

The speed and position of the particle swarm moving in the solution space are determined according to the following formula:

$$v_{t+1} = wv_t + c_1 r_1 (P_t - x_t) + c_2 r_2 (G_t - x_t) \tag{20}$$

$$x_{t+1} = x_t + v_{t+1} \tag{21}$$

where $x$ and $v$ represent the position and velocity of particles, respectively; $w$ is the acceleration factor; $c_1$ or $c_2$ is the acceleration constant; and $r_1$ or $r_2$ is a random number between 0 and 1.

The implementation process of PSO for PI parameters is shown in Figure 11.

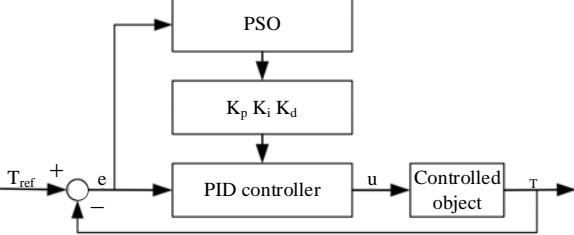

**Figure 11.** Schematic diagram of PSO PI implementation process.

The optimization problem of PI controller parameter tuning is the need to determine a group of appropriate parameters, namely $K_p$ and $K_i$, so that the control output index can reach the optimum value. Common error performance indicators include ISE, IAE, ITAE, etc. [45]. This paper selects ITAE indicators and defines the fitness function as shown in Formula (22):

$$J = \int_0^\infty t|e(t)|\mathrm{d}t \tag{22}$$

For PI control, PSO generates a group of two-dimensional particle swarms. The position attribute $x_i = (x_{i1}, x_{i2})$ for particle $i$. When their fitness value $J_i$ meets the optimal conditions, $x_{i1}$ and $x_{i2}$ are assigned to $K_p$ and $K_i$ parameters, respectively. Therefore, the process of PSO is the process of PI controller parameter tuning [44].

For the DC microgrid system outlined in this paper, the PSO algorithm is used to simultaneously self-tune the PI controller parameters at each generation unit side, and the difference between the actual output power of each generation unit and the reference value of the power output through droop control is summed as the error performance index (ITAE). In order to speed up the calculation and avoid the impact of the dimension disaster, this paper proposes a sequential optimization method for the three different types of power generation units involved in this paper: firstly, the PSO algorithm is used to optimize the PI parameters of the first type of power generation units, and the traditional PI control is used for the rest of power generation units. Then, the optimized PI parameters are used for the optimized units, and the PSO algorithm is used to optimize the PI parameters of the second type of power generation units, while the traditional PI control is used for other power generation units. Finally, the PI parameters of all cells are optimized in turn.

The specific optimization steps for each type of power generation unit are as follows:

(1) Initialize. Set parameters such as population size, iteration times and boundary conditions, and initialize the position, speed and fitness of particles.
(2) Assignment. The generated population particles are assigned to the proportional and integral variables of the PI controller of each generation unit in turn, and the model output error performance index is run.
(3) Judgment. Judge whether the algorithm reaches the set number of iterations or whether the ITAE value of the simulation model output is less than the set minimum fitness value. If the number of algorithm iterations is greater than the set number or the ITAE value is less than the set minimum fitness value, the algorithm is terminated. If the end condition of the algorithm is not met, each particle in the particle population will return to (2) after updating its position and speed and will exit the cycle until the exit condition of the algorithm is met.

The sample settings of the optimization algorithm in this paper are shown in Table 3.

**Table 3.** Main parameters of PSO algorithm.

| Inertia Weight | Acceleration Factor | Dimension | Population Size | Iterations | Minimum Fitness Value |
|---|---|---|---|---|---|
| $\omega_{max} = 0.9$ | $c_1 = 0.8$ | Dim = 12 | Size = 10 | Iter = 5 | Minfit = 0.001 |
| $\omega_{min} = 0.4$ | $c_2 = 0.5$ | Dim = 12 | Size = 10 | Iter = 5 | Minfit = 0.001 |

## 4. Simulation Test

### 4.1. Simulation System and Parameters

In this paper, the DC microgrid described in Figure 1 is built in the simulation software, including two groups of photovoltaic power generation units, three diesel generators, two groups of battery energy storage units, and one group of AA-CAES units. The main parameters of the system are shown in Table 4.

**Table 4.** Main parameters of the system.

| Parameter Name | Numerical Value | Parameter Name | Numerical Value |
|---|---|---|---|
| Rated voltage of DC bus (V) | 400 | Maximum power of photovoltaic unit (VA) | $1 \times 10^5$ |
| Rated power of diesel generator (VA) | $1 \times 10^6$ | Rated power of diesel generator (VA) | $1.3 \times 10^6$ |
| Rated speed of diesel generator (rpm) | 1500 | AA-CAES rated power (VA) | $1 \times 10^6$ |
| Rated power of storage battery (VA) | 6000 | Battery energy storage limit power (VA) | $1 \times 10^4$ |

Key parameters of main components of the AA-CAES system are shown in Table 5.

**Table 5.** Key parameters of main components of the AA-CAES system.

| Parameter Name | Numerical Value | Parameter Name | Numerical Value |
|---|---|---|---|
| Ambient air pressure (bar) | 1 | Number of heat exchange tubes (m) | 507 |
| Ambient temperature (K) | 298 | Inner diameter of pipe body (m) | 0.8 |
| Asynchronous motor | | Piston compressor | |
| Stator resistance ($\Omega$) | 0.0138 | Initial air pressure in cylinder (bar) | 1.011 |
| Stator leakage resistance (H) | 0.0002 | Stroke of each cylinder (m) | 0.3690 |
| Rotor resistance ($\Omega$) | 0.0077 | Crank radius of each cylinder (m) | 0.1845 |
| Rotor leakage inductance (H) | 0.0002 | Length of connecting rod of each cylinder (m) | 0.4716 |
| Excitation inductance (H) | 0.0077 | First stage: cylinder diameter (m) | 1.0868 |
| Moment of inertia (kg/m$^2$) | 2.9 | First stage: clearance volume (m$^3$) | 0.0012 |
| Centripetal turbine | | Second stage: cylinder diameter (m) | 0.5895 |
| Rotor inlet angle (degree) | 12.6 | Second stage: clearance volume (m$^3$) | 0.0006 |
| Rotor outlet angle (degree) | 33.6 | Third stage: cylinder diameter (m) | 0.0289 |
| First stage: rotor inlet diameter (m) | 0.1728 | Third stage: clearance volume (m$^3$) | 0.00035 |
| First stage: average diameter of rotor outlet (m) | 0.0773 | Gas cylinder | |
| First stage: rotor inlet width (m) | 0.0044 | Volume of air tank (m$^3$) | 760 |
| First stage: rotor outlet width (m) | 0.0148 | Initial air pressure (bar) | 60 |
| Second stage: rotor inlet diameter (m) | 0.2300 | Initial air temperature (K) | 298 |
| Second stage: average diameter of rotor outlet (m) | 0.1050 | Natural heat transfer coefficient | 2.24 |
| Second stage: rotor inlet width (m) | 0.0074 | Forced heat transfer coefficient | 10.52 |
| Second stage: rotor outlet width (m) | 0.0253 | Synchronous motor | |
| Third stage: rotor inlet diameter (m) | 0.2600 | Rated power (VA) | $1 \times 10^6$ |
| Third stage: average diameter of rotor outlet (m) | 0.1144 | Rated line voltage (V) | 400 |
| Third stage: rotor inlet width (m) | 0.0192 | Stator resistance (pu) | 0.0095 |
| Third stage: rotor outlet width (m) | 0.0642 | Leakage inductance of stator winding (pu) | 0.05 |
| Shell and tube heat exchanger | | $d$-axis excitation inductance (pu) | 2.06 |
| Length of heat exchanger tube (m) | 4.2 | $q$-axis excitation inductance (pu) | 1.51 |
| Outer diameter of heat exchanger tube (m) | 0.0250 | Excitation winding resistance (pu) | 0.0020 |
| Inner diameter of heat exchanger tube (m) | 0.0225 | Excitation winding inductance (pu) | 0.0034 |

### *4.2. Example Analysis of the Effect of Composite Energy Storage on DC Microgrid Voltage Stabilization*

In this section, the simulation analysis is mainly conducted on the DC bus voltage fluctuation when there is no AA-CAES system connected to the DC microgrid, that is, in relation to whether the composite energy storage is used, in order to verify the positive role of the composite energy storage system composed of AA-CAES and batteries in participating in the DC microgrid voltage stabilization control process.

### 4.2.1. Light Intensity Fluctuation

(1)    Example 1

As shown in Figure 12, the fluctuation of light intensity in Example 1, simulates the situation where the light intensity changes in a short time and then remains constant for a period of time.

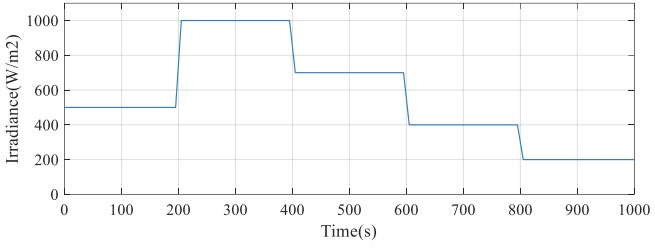

**Figure 12.** Fluctuation of light intensity in Example 1.

The output voltage value of the constant current bus in Example 1 is shown in Figure 13.

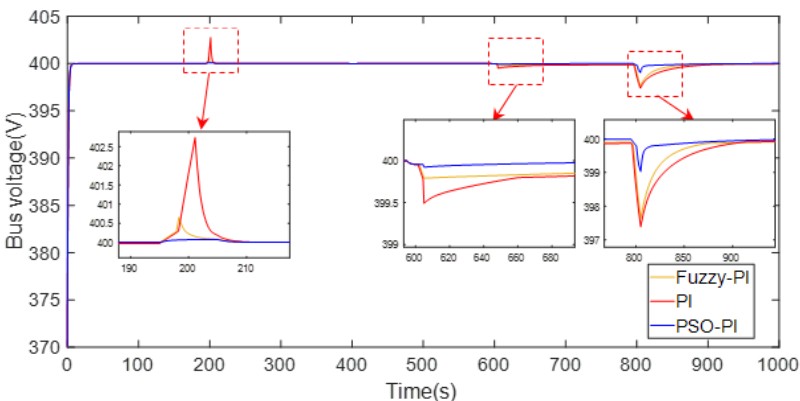

**Figure 13.** Bus voltage in Example 1.

It can be seen from Figure 13 that when the light intensity changes greatly, the bus voltage fluctuates in a small range. When PI control is adopted, the maximum fluctuation is 402.643 V, $1.0066v^*$, $\Delta v = 0.66\%v^*$. When using Fuzzy PI control, the maximum fluctuation is 397.637 V, $0.9941v^*$, $\Delta v = 0.59\%v^*$. When PSO PI control is adopted, the maximum fluctuation is 399.026 V, $0.9976v^*$, $\Delta v = 0.24\%v^*$. From the simulation results, it can be seen that the adopted control strategy can maintain the DC bus voltage stability. When disturbed, the disturbance is within the allowable fluctuation range of voltage ($-10\%v^* \sim +10\%v^*$), and the bus voltage can quickly return to the rated value. PSO PI control has the best effect, followed by fuzzy PI control, but both are better than traditional PI control.

When the AA-CAES system in the DC microgrid is removed, the DC bus output voltage of this scenario, Example 1, is shown in Figure 14.

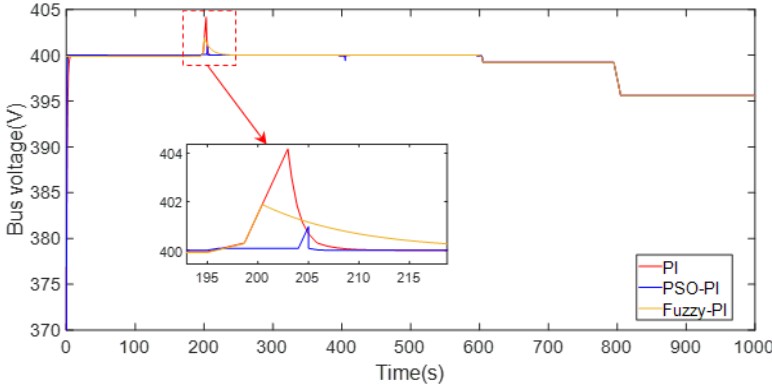

**Figure 14.** Bus voltage without AA-CAES in Example 1.

It can be seen from Figure 14 that when the light intensity suddenly increases, the bus voltage fluctuates in a small range, and that when PSO PI control is used for parameter optimization, the bus voltage fluctuation is the smallest, followed by fuzzy PI control, and the traditional PI control is the worst. After 600 s, the light intensity is too low, the power generated by each unit in the DC microgrid is insufficient to support the bus voltage, and the bus voltage drops below the rated value.

By comparing Figures 13 and 14, AA-CAES system can be seen to have the advantages of fast response and large capacity, being able to effectively deal with the instability of renewable energy, and of playing a positive role in maintaining the stability of DC bus voltage.

The output power of the two groups of photovoltaic power generation units in Example 1 is shown in Figure 15.

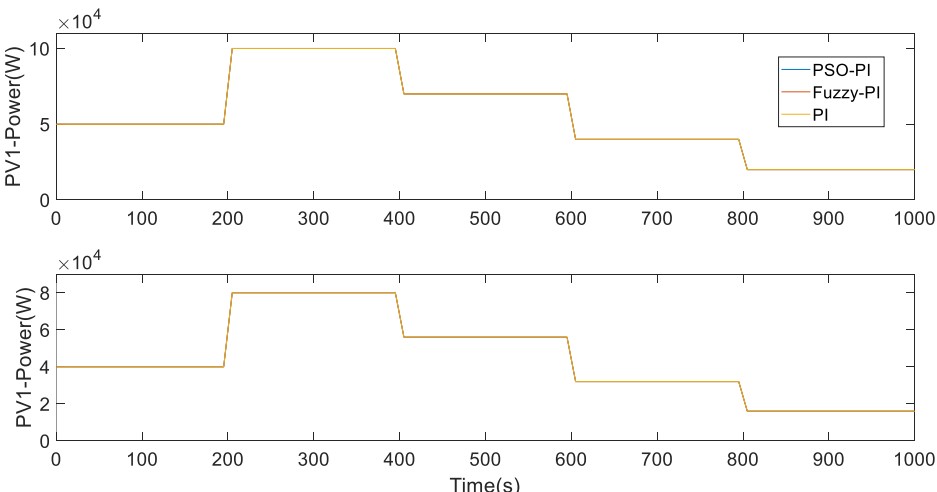

**Figure 15.** Output power of photovoltaic unit in Example 1.

The two groups of photovoltaic power generation units are controlled by MPPT. As can be seen from Figure 15, the two groups of photovoltaic power generation units can track the change in light intensity and always work at the maximum power point to maximize the utilization of photovoltaic energy.

The output power of three groups of diesel generators, outlined in Example 1, is shown in Figure 16.

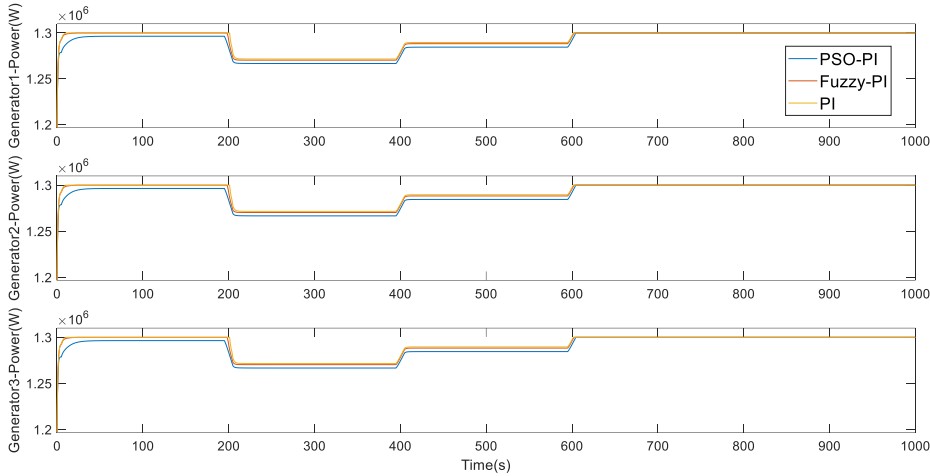

**Figure 16.** Output power of diesel generator in Example 1.

It can be seen from Figure 16 that the output power of the three groups of diesel generators is complementary to that of the photovoltaic unit. That is, when the light intensity is high, the output power is low, and when the light intensity is low, the output power is high. After 600 s, the light intensity is too low and the diesel generator reaches the output limit. In order to maintain the stability of the DC bus voltage, the maximum power is continuously output.

The output power of the two groups of battery energy storage units in Example 1 is shown in Figure 17.

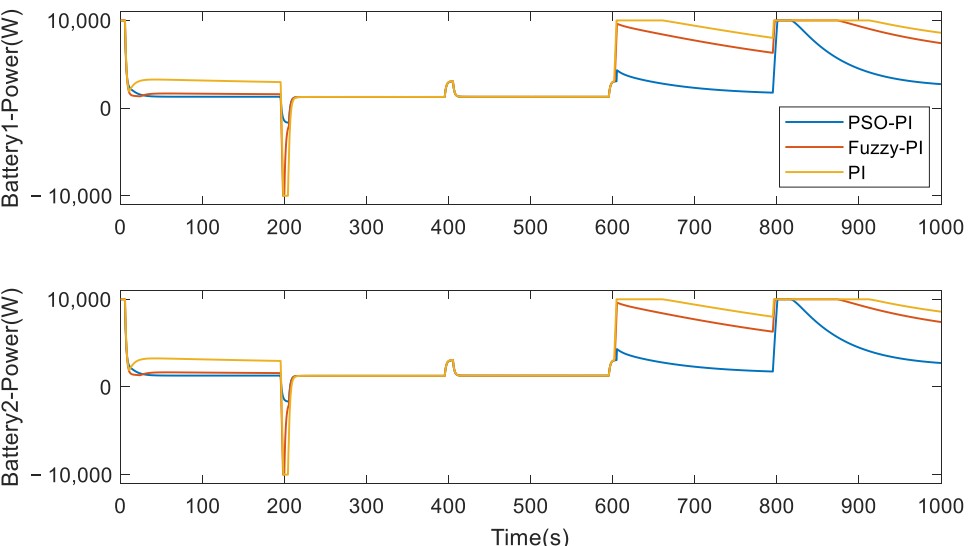

**Figure 17.** Output power of battery in Example 1.

The battery energy storage unit has the characteristic of fast response but has low energy density. It can be seen from Figure 17 that when the output power of other units in the DC microgrid is insufficient to support the bus voltage, the battery energy storage unit responds quickly and increases the output power to maintain the bus voltage stability. In order to reduce the life loss of the battery energy storage unit, its overcharge and overdischarge should be avoided. From the simulation results, when PSO PI control is adopted, the battery energy storage unit works at its limited output power for the shortest time, with the best protection effect for the energy storage unit, followed by fuzzy PI control, and with the traditional PI control being the worst.

The output power of AA-CAES in Example 1 is shown in Figure 18.

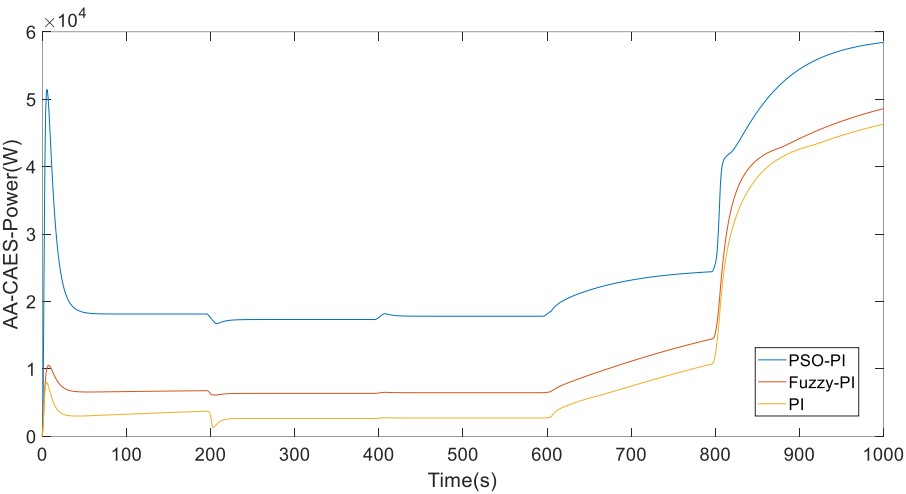

**Figure 18.** AA-CAES output power in Example 1.

AA-CAES technology has the characteristic of large capacity, but its climbing speed is slower than that of battery energy storage unit. When the power from other units in the DC microgrid is insufficient to support the bus voltage, AA-CAES adjusts the output to maintain the bus voltage at the rated voltage level. It can be seen from Figure 18 that when PSO PI control is adopted, AA-CAES has the largest output power, which can reduce the output of diesel generator sets and the use of diesel fuel resources. Moreover, it has the fastest climbing speed and can recover bus voltage more quickly. The fuzzy PI control effect is the second fast, and the traditional PI control effect has the worst ability.

(2)  Example 2

The fluctuation in light intensity in Example 2 is shown in Figure 19, which simulates the continuous change in light intensity in reality.

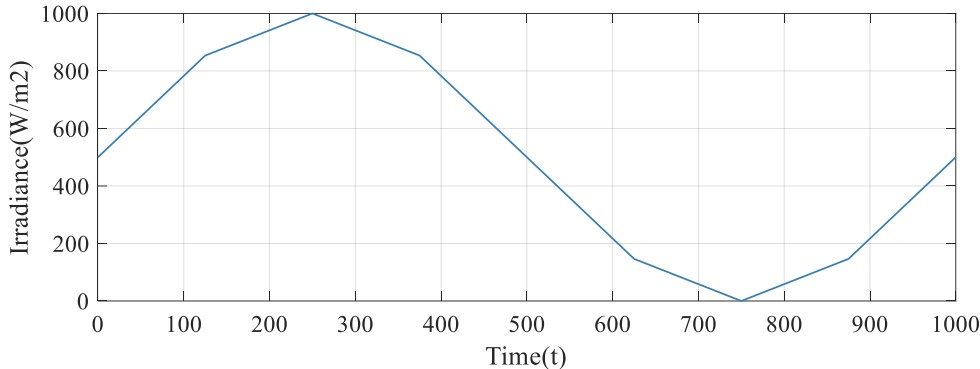

**Figure 19.** Fluctuation in light intensity in Example 2.

The output voltage of DC bus in Example 2 is shown in Figure 20.

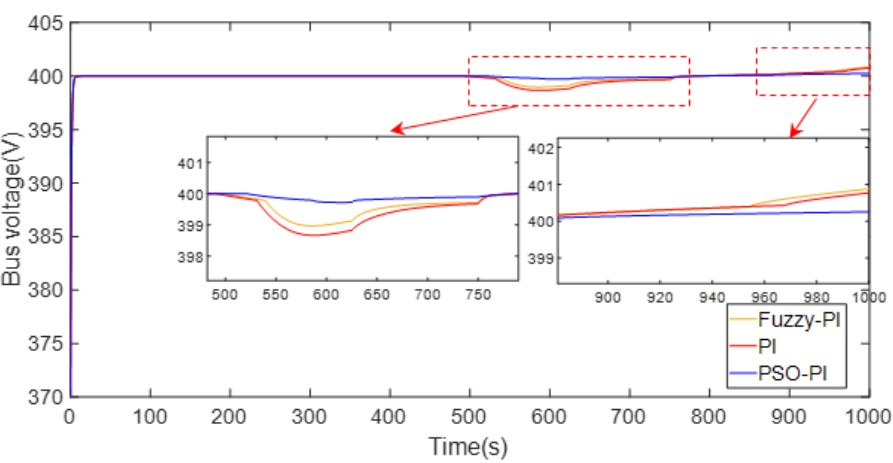

**Figure 20.** Fluctuation of light intensity in Example 2.

It can be seen from Figure 20 that when the light intensity is small, the bus voltage fluctuates in a small range. When PI control is adopted, the maximum fluctuation is 398.665 V, $0.9967v^*$, $\Delta v = 0.33\%v^*$. When using fuzzy PI control, the maximum fluctuation is 398.970 V, $0.9974v^*$, $\Delta v = 0.26\%v^*$. When PSO PI control is adopted, the maximum fluctuation is 399.721 V, $0.9993v^*$, $\Delta v = 0.07\%v^*$. From the simulation results, it can be seen that the adopted control strategy can maintain the DC bus voltage stability. When the light intensity is low, the disturbance is within the allowable voltage fluctuation range ($-10\%v^* \sim +10\%v^*$), and the bus voltage can quickly return to the rated value. PSO PI control has the best effect, followed by fuzzy PI control, but both are better than traditional PI control methods.

When the AA-CAES system in the DC microgrid is removed, the DC bus output voltage value of Example 2 is shown in Figure 21.

According to the comparison between Figures 20 and 21, when the light intensity is too low, the bus voltage of the DC microgrid system with AA-CAES fluctuates and can return to the rated voltage. After removing AA-CAES, the DC bus voltage fluctuates greatly and cannot return to the rated value.

The output power of the two groups of photovoltaic power generation units in Example 2 is shown in Figure 22.

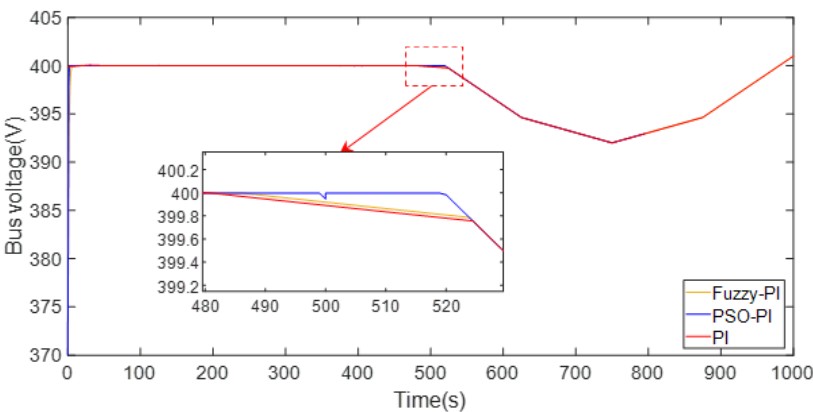

**Figure 21.** Bus voltage in Example 2 without AA-CAES.

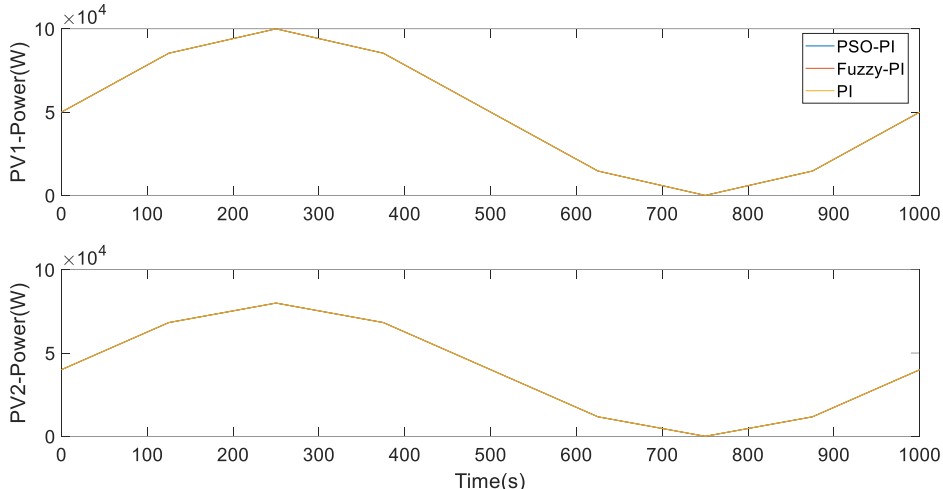

**Figure 22.** Output power of photovoltaic power generation unit in Example 2.

The two groups of photovoltaic power generation units are controlled by MPPT. As can be seen from Figure 22, the two groups of photovoltaic power generation units can track the change in light intensity and always work at the maximum power point in order to maximize the utilization of photovoltaic energy.

The output power of the three groups of diesel generators in Example 2 is shown in Figure 23.

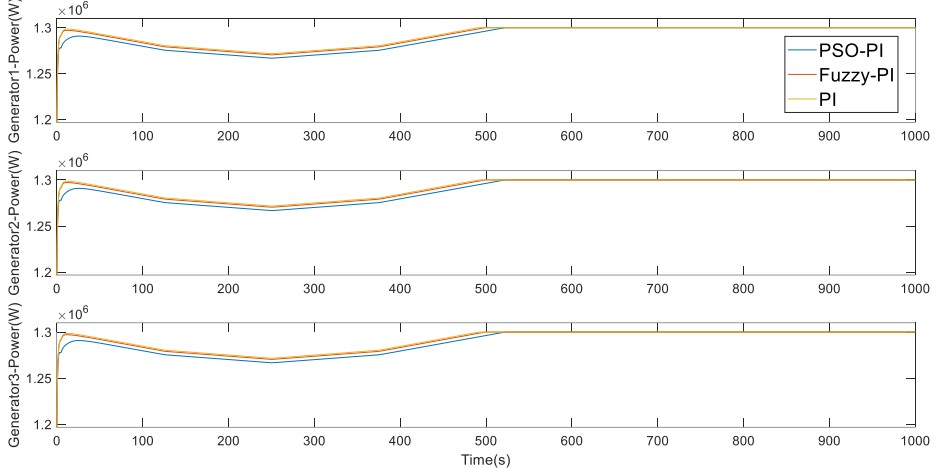

**Figure 23.** Output power of diesel generator in Example 2.

It can be seen from Figure 23 that the output power of the three groups of diesel generators is complementary to that of the photovoltaic unit. After 500 s, the light intensity is too low and the diesel generator reaches the output limit. In order to maintain the stability of the DC bus voltage, the maximum power is continuously output.

The output power of the two groups of battery energy storage units in Example 2 is shown in Figure 24.

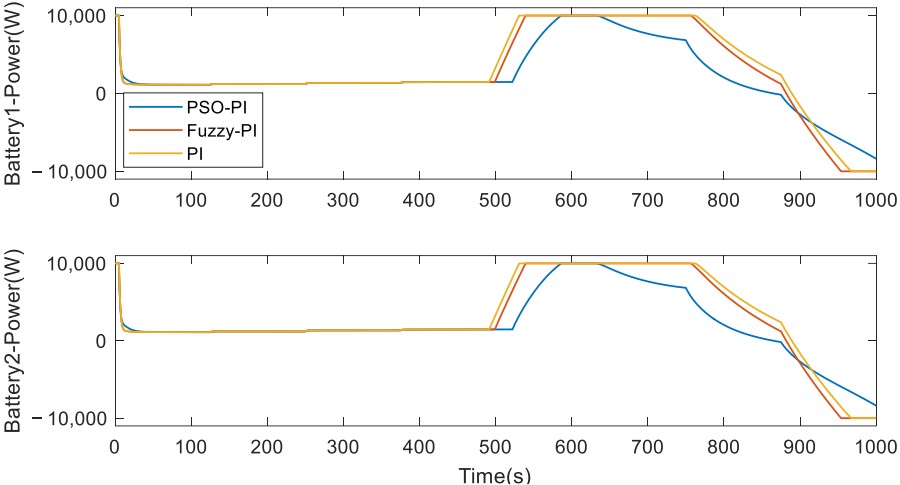

**Figure 24.** Output power of battery in Example 2.

It can be seen from Figure 24 that when the output power of other units in the DC microgrid is insufficient to support the bus voltage, the battery energy storage unit responds quickly and increases the output power to maintain the bus voltage stability. In addition, when PSO PI control is adopted, the battery energy storage unit has the shortest time to work at its limited output power and has the best protection effect on the energy storage unit, followed by fuzzy PI control, and the traditional PI control has the worst effect.

The output power of AA-CAES in Example 2 is shown in Figure 25.

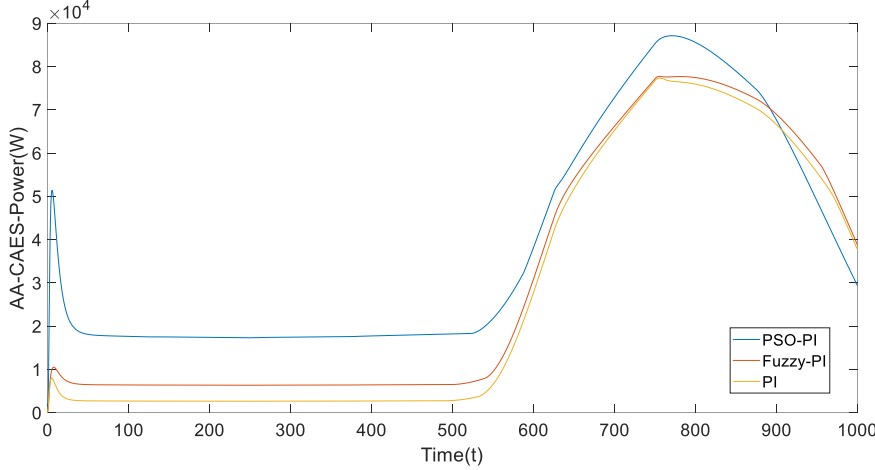

**Figure 25.** Output power of AA-CAES in Example 2.

In this example, after 500 s, the light intensity is low, and the power from other units in the DC microgrid is insufficient to support the bus voltage. AA-CAES adjusts the output to maintain the bus voltage at the rated voltage. It can be seen from Figure 25 that when PSO PI control is adopted, AA-CAES has the largest output power, which can reduce the output of diesel generator sets and the use of diesel resources. Moreover, after 628 s, the climbing speed is the fastest, which can recover bus voltage faster. The fuzzy PI control effect is the second fastest, and the traditional PI control effect is the worst.

(3)　Example 3

The fluctuation of light intensity in Example 3 is shown in Figure 26, simulating the situation where the light intensity first increases and then decreases in a day.

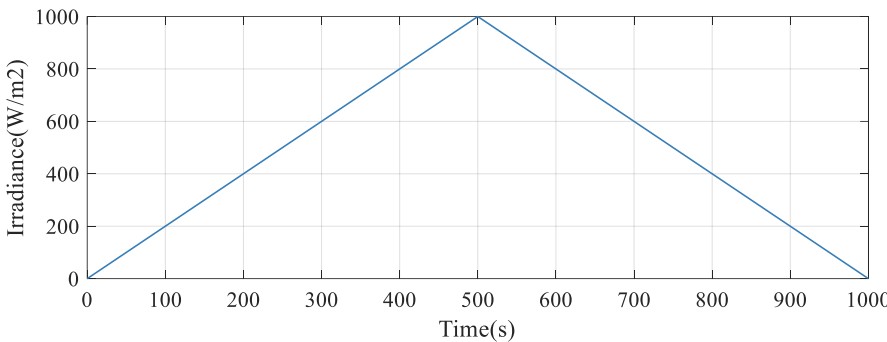

**Figure 26.** Fluctuation of light intensity in Example 3.

The DC bus output voltage value of Example 3 is shown in Figure 27.

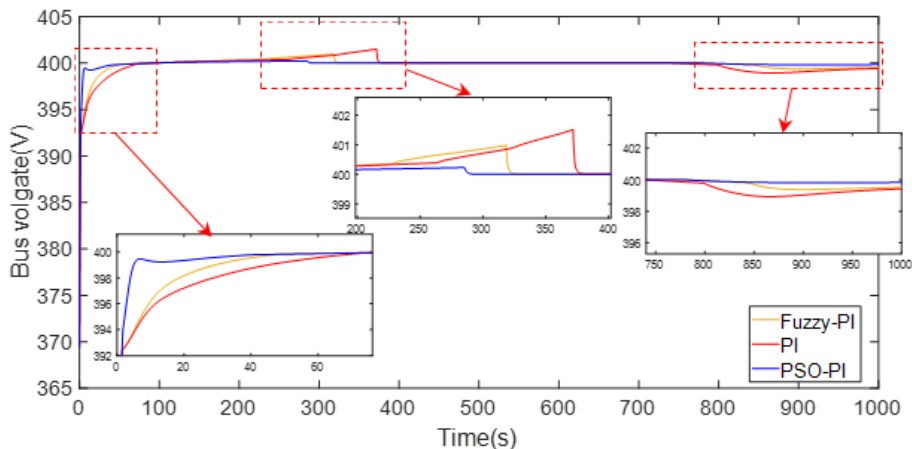

**Figure 27.** Bus voltage in Example 3.

It can be seen from Figure 27 that the bus voltage fluctuates in a small range during the gradual increase and decrease in light intensity. When PI control is adopted, the maximum fluctuation is 401.512 V, $1.0038v^*$, $\Delta v = 0.38\%v^*$. When using Fuzzy PI control, the maximum fluctuation is 400.981 V, $1.0025v^*$, $\Delta v = 0.25\%v^*$. When PSO-PI control is adopted, the maximum fluctuation is 400.229 V, $1.0006v^*$, $\Delta v = 0.06\%v^*$. From the simulation results, it can be seen that the adopted control strategy can maintain the stability of DC bus voltage. When the light intensity changes, the disturbance is within the allowable fluctuation range of voltage ($-10\%v^* \sim +10\%v^*$), and the bus voltage can quickly return to the rated value. When PSO PI control is used, the bus voltage climbs to the rated value at the startup stage at the fastest speed, with the best control effect, followed by fuzzy PI control, and the traditional PI control is the worst method.

The DC bus output voltage of the AA-CAES system in which the DC microgrid is removed, as described in Example 3, is shown in Figure 28.

Through the comparison between Figures 27 and 28, it is easy to draw the conclusion that AA-CAES system plays an important role in DC micro grid voltage stabilization control, which is similar to what was shown Example 1 and Example 2.

The output power of the two groups of photovoltaic power generation units in Example 3 is shown in Figure 29.

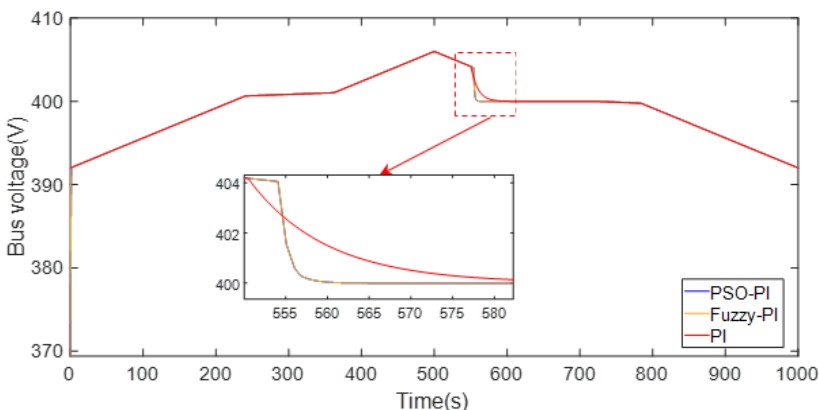

**Figure 28.** Bus voltage in Example 3 without AA-CAES.

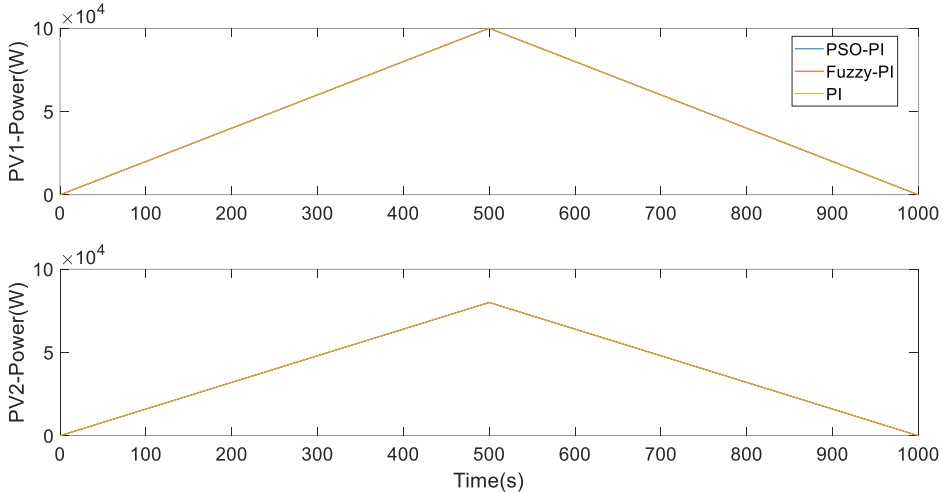

**Figure 29.** Output power of photovoltaic power generation unit in Example 3.

The two groups of photovoltaic power generation units are controlled by MPPT. As can be seen from Figure 29, the two groups of photovoltaic power generation units can track the change in light intensity and always work at the maximum power point to maximize the utilization of photovoltaic energy.

The output power of the three groups of diesel generators in Example 3 is shown in Figure 30.

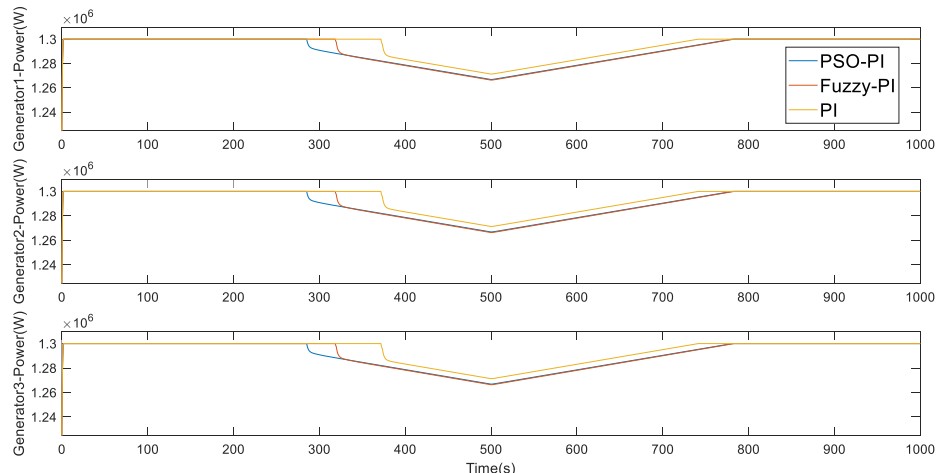

**Figure 30.** Output power of diesel generator in Example 3.

It can be seen from Figure 30 that the output power of the three groups of diesel generators is complementary to the output power of the photovoltaic unit. When the light intensity is too low, in order to maintain the DC bus voltage stability, the diesel generator reaches the output limit and continuously outputs the maximum power.

The output power of the two battery energy storage units in Example 3 is shown in Figure 31.

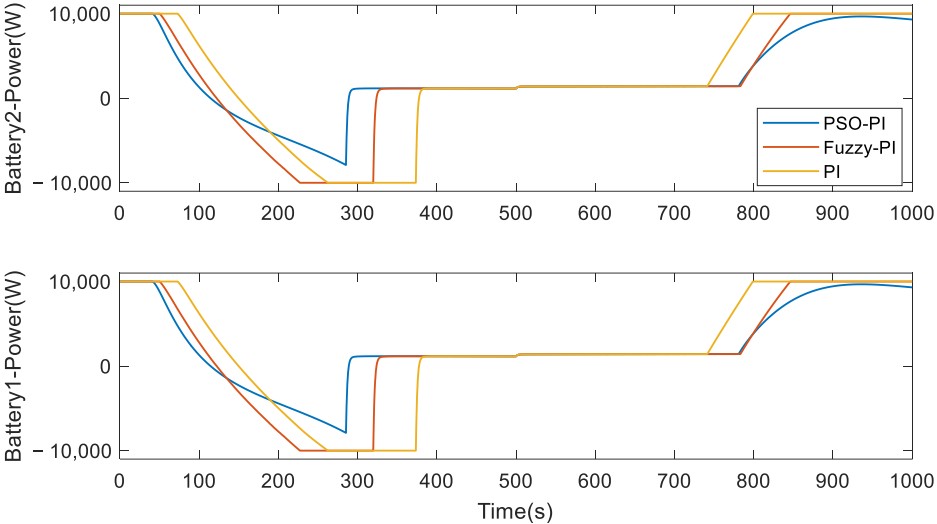

**Figure 31.** Output power of battery in Example 3.

It can be seen from Figure 31 that when the bus voltage is higher than the rated value, the battery energy storage unit will enter the charging state. When the output power of other units in the DC microgrid is insufficient to support the bus voltage, the battery energy storage unit responds quickly and increases the output power to maintain the bus voltage stability. When using PSO PI control, the battery energy storage unit has the shortest time to work at its limit value, the best protection effect for the energy storage unit, followed by fuzzy PI control, and the traditional PI control has the worst effect.

The output power of AA-CAES in Example 3 is shown in Figure 32.

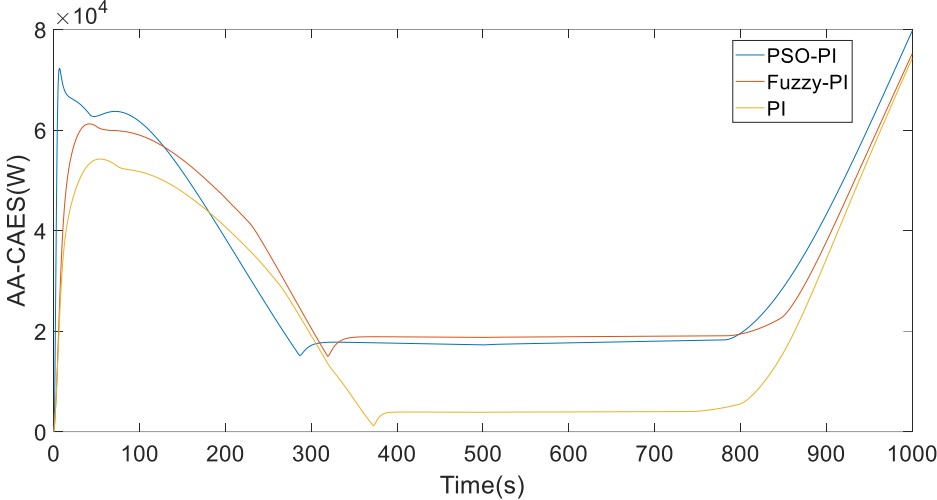

**Figure 32.** Output power of AA-CAES in Example 3.

In this example, the light intensity is low after 0 ~ 300 s and 800 s, and the power from other units in the DC microgrid is insufficient to support the bus voltage. AA-CAES adjusts the output to maintain the bus voltage to the rated voltage. As can be seen from Figure 32, when the light intensity is low during the startup phase and after 800 s and

when PSO PI control is used, AA-CAES has the largest output power and can stabilize the DC bus voltage faster, followed by fuzzy PI control, and the traditional PI control has the worst effect.

### 4.2.2. Power Generation Unit Failure

In this example, when the simulated light intensity is unchanged, a diesel generator stops suddenly due to a fault. Diesel generator 3 is designed to stop the operation in 200 s due to fault.

The output voltage of the DC bus is shown in Figure 33.

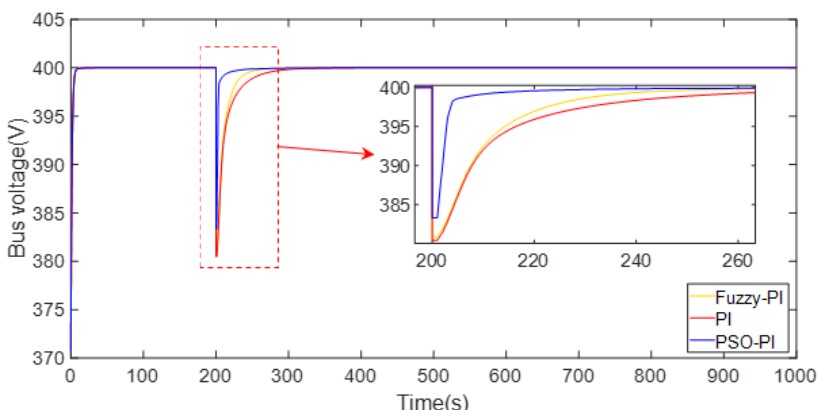

**Figure 33.** DC bus voltage when one generator fails.

It can be seen from Figure 33 that when a generator stops running, the bus voltage fluctuates within a small range. When PI control is adopted, the fluctuation value is 380.461 V, $0.9512v^*$, $\Delta v = 4.88\%v^*$. When using Fuzzy PI control, the maximum fluctuation is 380.899 V, $0.9522v^*$, $\Delta v = 4.78\%v^*$. When PSO PI control is adopted, the maximum fluctuation is 383.331 V, $0.9583v^*$, $\Delta v = 4.17\%v^*$. It can be seen from the simulation results that the adopted control strategy can maintain the stability of DC bus voltage. When a generator stops running, the disturbance is within the allowable fluctuation range of voltage $(-10\%v^* \sim +10\%v^*)$ and traditional PI control has the worst effect.

When the AA-CAES system in the DC microgrid is removed, the output voltage of the DC bus changes, as is shown in Figure 34.

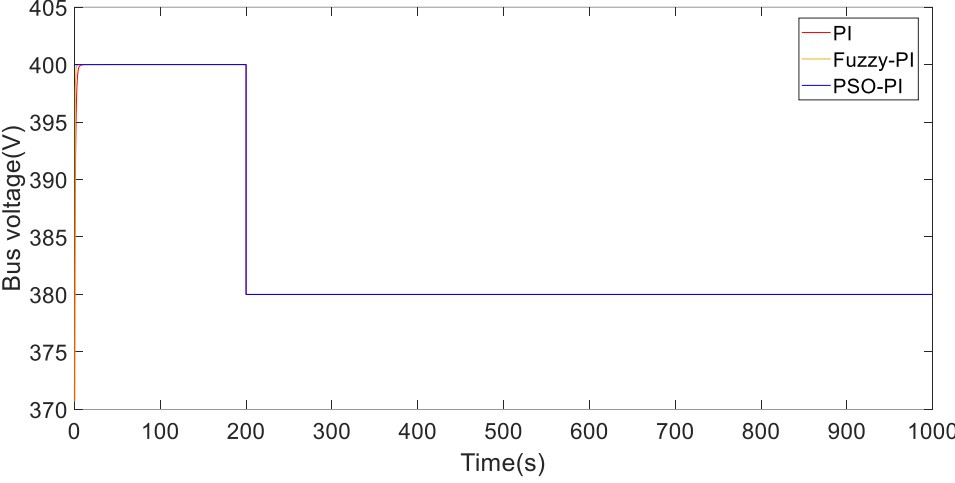

**Figure 34.** DC bus voltage when one generator fails without AA-CAES.

In Figure 34, due to the failure of one generator, the output power of the remaining generation unit is insufficient to support the bus voltage, and the bus voltage drops to 380 V and cannot be restored to the rated value. By comparing Figures 33 and 34, we can

obtain the same conditions as the light intensity fluctuation calculation example, that is, AA-CAES system can participate in the DC microgrid voltage regulation process and play an important role in the DC microgrid voltage stabilization control.

The output power of the two groups of photovoltaic power generation units is shown in Figure 35.

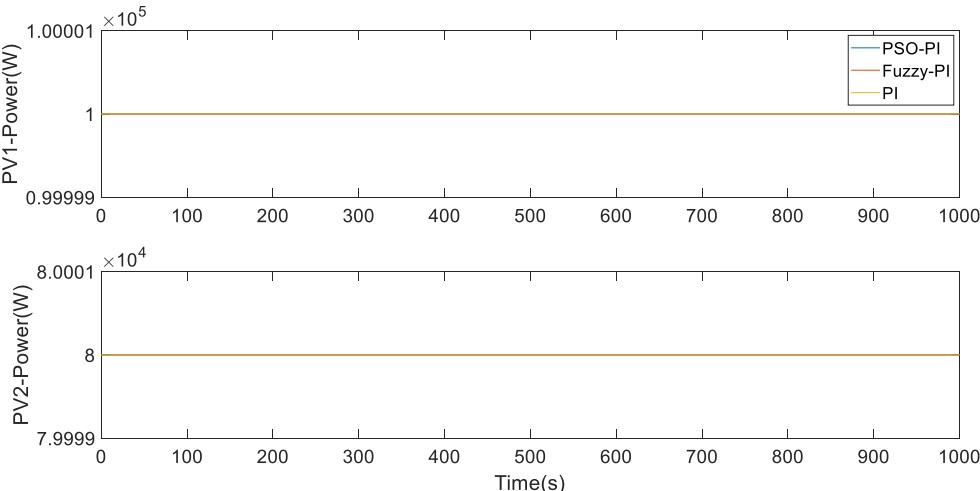

**Figure 35.** Output power of photovoltaic power generation unit when one generator fails.

In this example, the light intensity is unchanged, and both groups of photovoltaic power generation units operate at the maximum power point.

The output power of the three groups of diesel generators is shown in Figure 36.

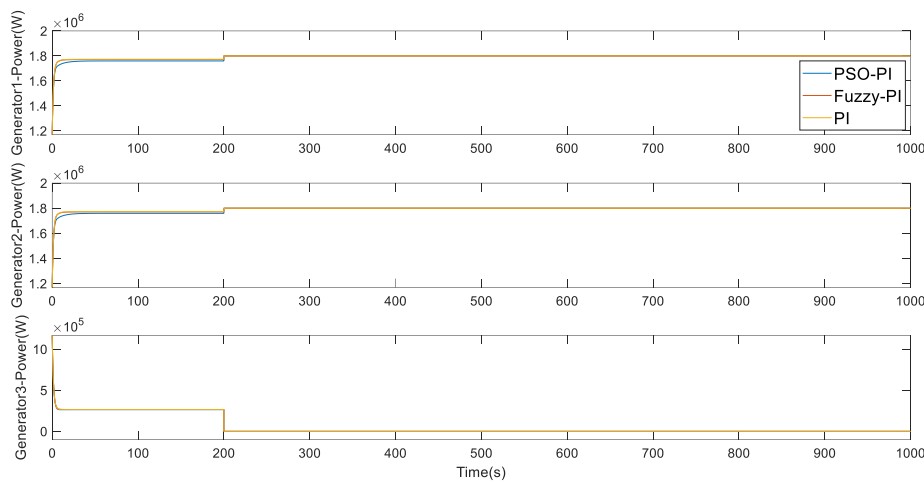

**Figure 36.** Output power of diesel generator when one generator fails.

It can be seen from Figure 36 that diesel generator 3 failed at 200 s and ceased operation. In order to maintain the stability of DC bus voltage, the other two generators rapidly increased their output and continuously outputted the maximum power when reaching the output limit.

The output power of the two battery energy storage units is shown in Figure 37.

It can be seen from Figure 37 that when the output power of other units in the DC microgrid is insufficient to support the bus voltage, the battery energy storage unit responds quickly and increases the output power to maintain the bus voltage stability. When PSO PI control is adopted, the battery energy storage unit works at its limit value for the shortest time and has the best protection effect on the energy storage unit, which is conducive to extending the service life of the energy storage unit. This method followed in terms of performance by fuzzy PI and traditional PI control.

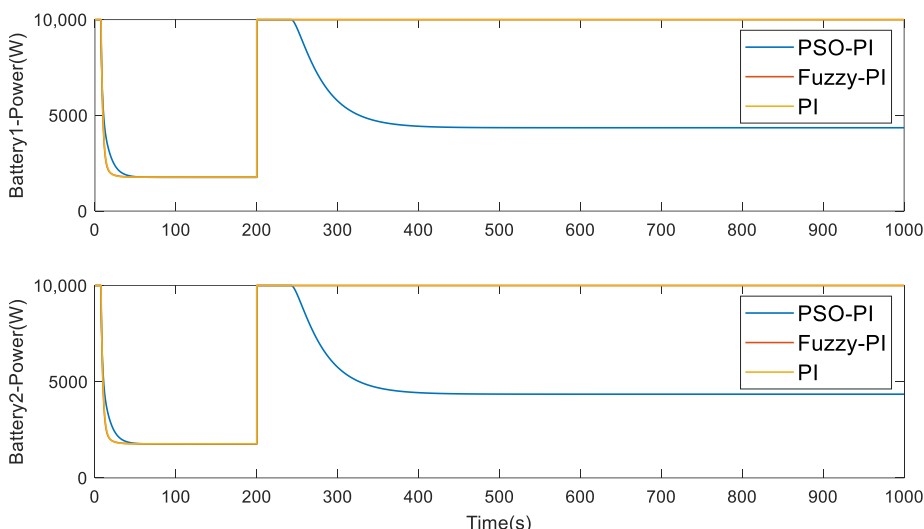

**Figure 37.** Output power of storage battery when one generator fails.

The output power of AA-CAES is shown in Figure 38.

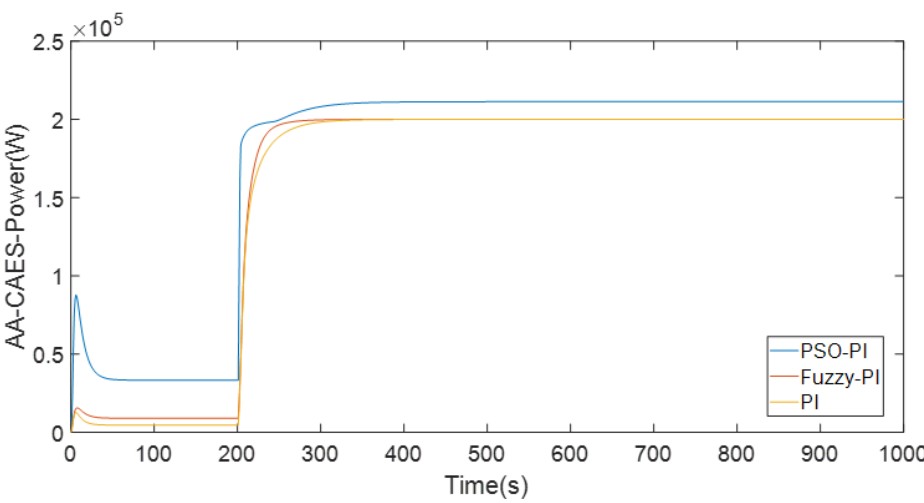

**Figure 38.** Output power of AA-CAES when one generator fails.

It can be seen from Figure 38 that when PSO PI control is adopted, AA-CAES has the largest output power before 200 s, which can reduce the output of diesel generator sets and reduce the use of diesel resources. After 200 s of generator failure, its climbing speed is the fastest, and it can restore the DC bus voltage the fastest, followed by fuzzy PI and traditional PI control.

*4.3. Composite Energy Storage Analysis*

4.3.1. AA-CAES Analysis

Based on the simulation module of key components of the AA-CAES system, this paper builds a megawatt-level AA-CAES full-system dynamic simulation model. The AA-CAES system structure diagram for simulation analysis is shown in Figure 39.

1.    Light intensity fluctuation

The simulation results of AA-CAES system are analyzed by using PSO PI control when the light intensity fluctuates.

Figure 40a shows the pressure changes in the three-stage cylinder of the piston compressor. Figure 40b shows the change in generator speed during power generation. Figure 40c shows the variation of generated power during power generation.

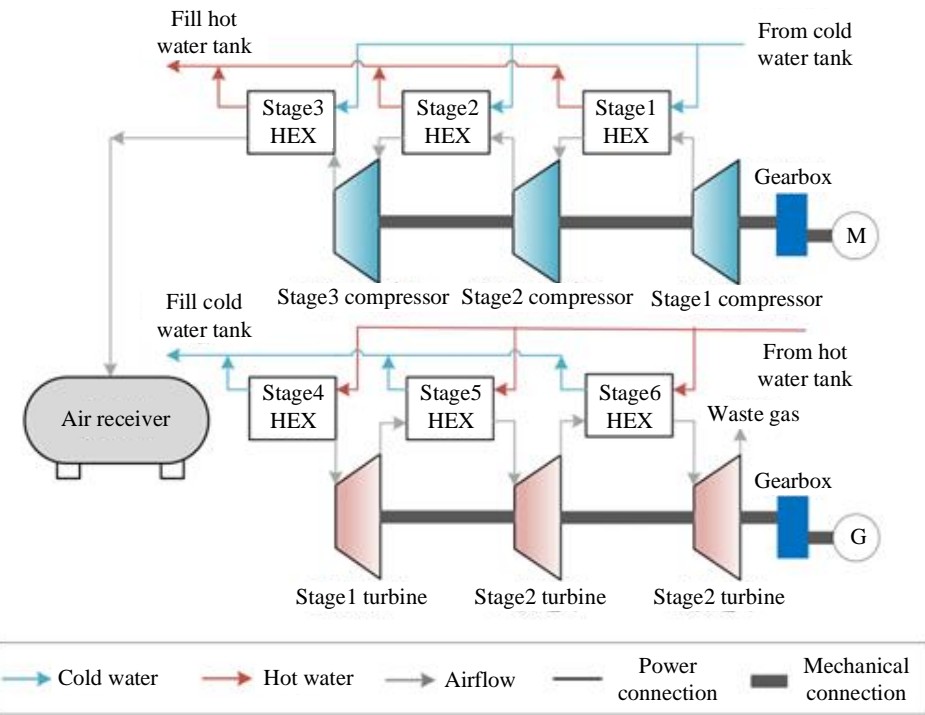

**Figure 39.** Schematic diagram of AA-CAES system structure for simulation.

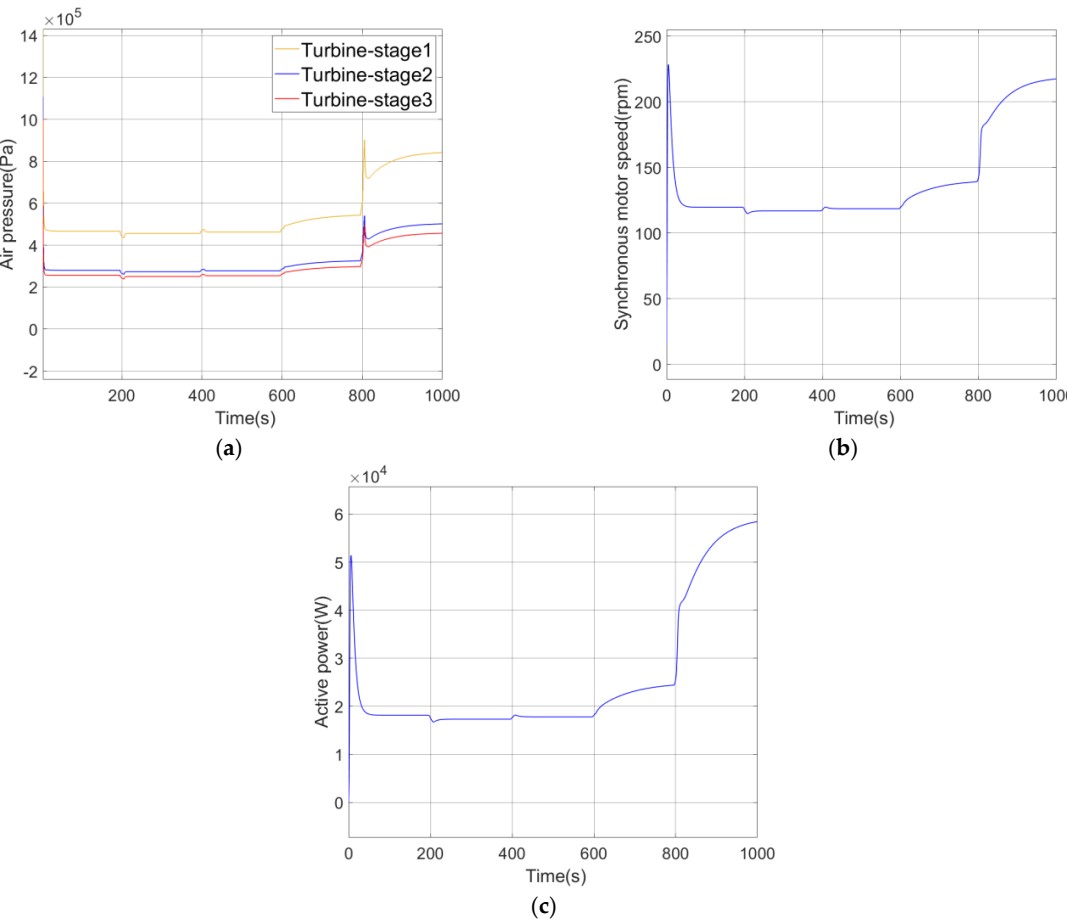

**Figure 40.** Simulation results of AA-CAES system participating in microgrid voltage regulation under light fluctuation: (**a**) Change in air pressure in cylinder of piston compressor; (**b**) Speed change in synchronous generator; (**c**) Change in power generated by the system.

The AA-CAES system dynamic simulation model constructed in this paper can effectively reflect the dynamic changes of physical quantities in the key components of the system over time. It can be seen from Figure 40 that the air pressure in the cylinder of the piston compressor, the speed of the synchronous generator and the power generated by the system change synchronously. When the power generated by other units of the microgrid is insufficient to support the bus voltage, researchers should increase the output and stabilize the bus voltage.

2.  Power generation unit failure

In this section, the simulation results of AA-CAES system are analyzed for an example of a generator failure when PSO PI control is used.

Figure 41a shows the pressure changes in the three-stage cylinder of the piston compressor. Figure 41b shows the change in generator speed during power generation. Figure 41c shows the variation in generated power during power generation.

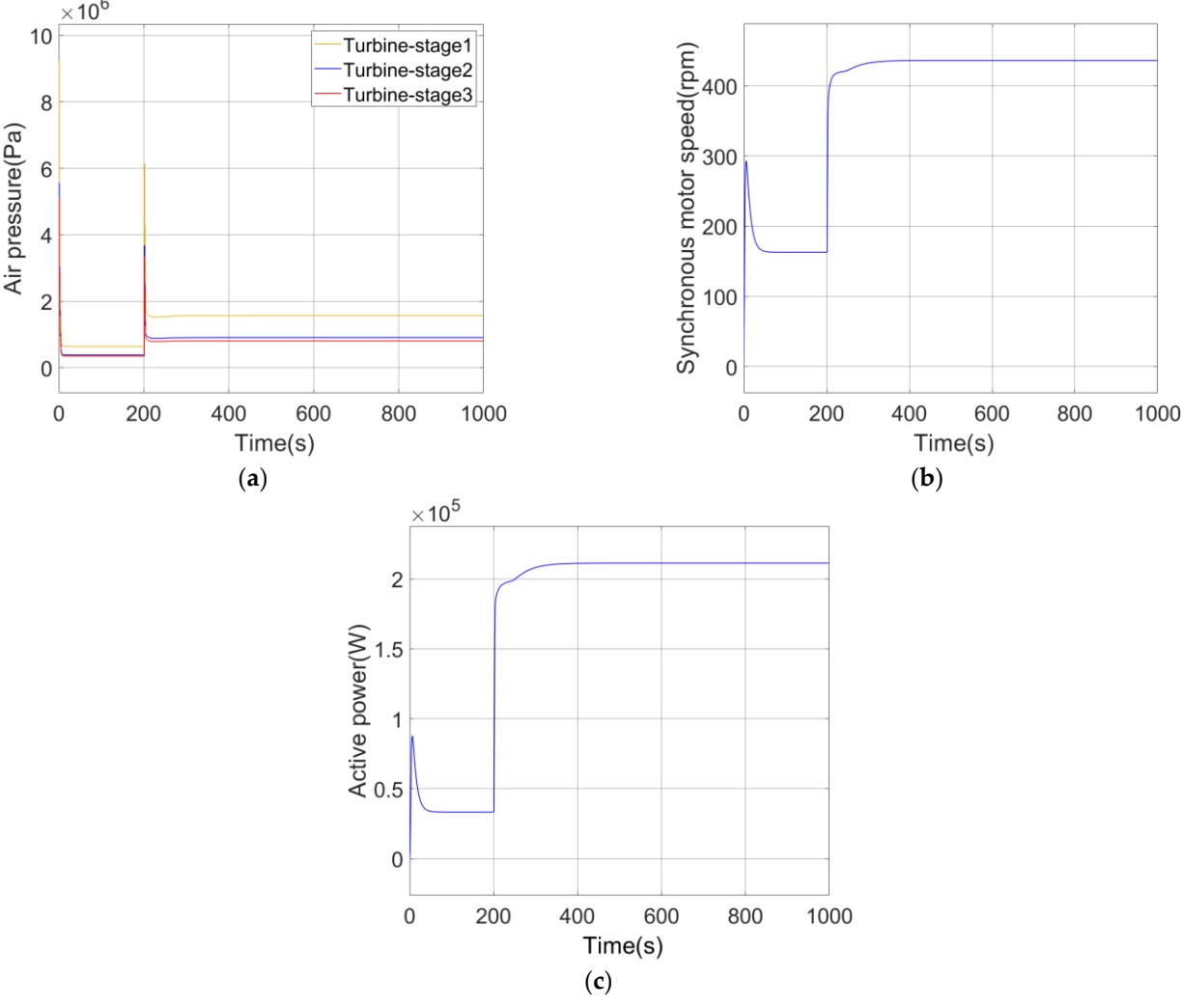

**Figure 41.** Simulation results of AA-CAES system participating in microgrid voltage regulation in case of a generator failure: (**a**) Change in air pressure in cylinder of piston compressor; (**b**) Speed change in synchronous generator; (**c**) Change in power generated by the system.

It can be seen from Figure 41 that after 200 s, the air pressure in the cylinder of the piston compressor increases, the synchronous generator speed increases, the system power generation is adjusted, the output is increased, and the bus voltage is stabilized.

### 4.3.2. Battery Analysis

This section mainly focuses on the simulation analysis of the SOC change in the battery with or without AA-CAES being connected to the DC microgrid under different calculation examples.

1. Light intensity fluctuation

Taking example 1, when the light intensity fluctuates, PSO PI control is adopted.

When AA-CAES is connected to the DC microgrid or not, the change in battery SOC value is shown in Figure 42, respectively.

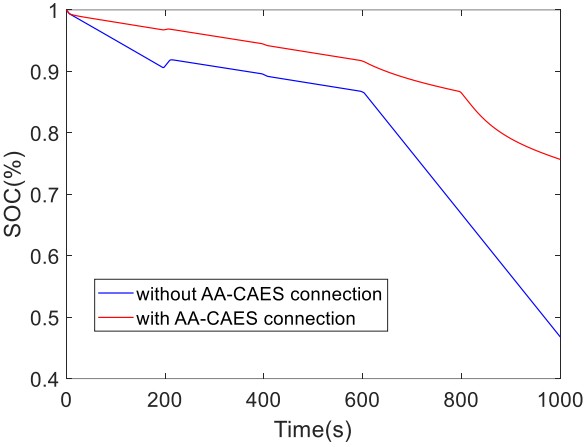

**Figure 42.** Change in SOC value of storage battery when light intensity fluctuates.

It can be seen from the simulation results in Figure 42 that without AA-CAES access, the battery SOC decreases faster and the charge–discharge cycle is shorter.

2. Power generation unit failure

This section addresses the calculation example of a generator failure when using PSO PI control.

When AA-CAES is connected to the DC microgrid or not, the change in battery SOC value is shown in Figure 43, respectively.

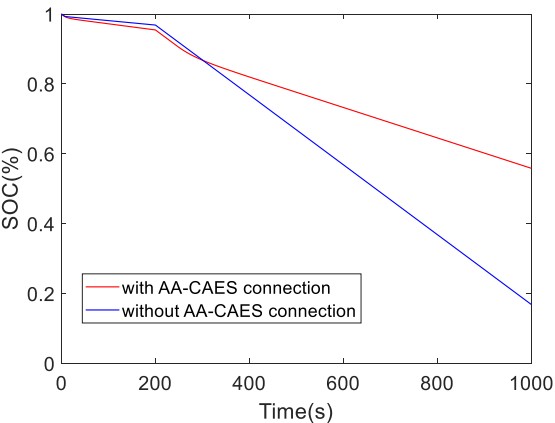

**Figure 43.** Change in battery SOC value when a generator fails.

It can be seen from Figure 43 that in the same time when AA-CAES is connected to the DC microgrid, the SOC value of the battery decreases more slowly. That is, when composite energy storage with an AA-CAES and battery is used, the charging and discharging cycle of the battery cell can be extended, thus increasing its service life. Thus, the correctness and effectiveness of the composite energy storage system for stabilizing the DC bus voltage and prolonging the service life of the battery are verified through numerical simulation.

It should be noted that the reliability assessment of the ESS has often been incorporated into power systems to consider thermally induced events, and the previous studies have generally highlighted the consideration of ESS application from the stability perspective, which has been ignored in this paper. Moreover, the degradation process is accelerated at high temperatures and some SOC ranges, resulting in a shortened lifespan of battery energy storage, and the service performance of the BESS is highly related to temperature and SOC values. Thus, the reliability assessment of the ESS should be considered in the future work of the DC microgrid control research.

## 5. Conclusions

In this paper, a composite energy storage system consisting of advanced adiabatic compressed air energy storage and storage battery, based on droop control and improved PI control, is proposed for use in the DC microgrid voltage stabilization control process. The AA-CAES has the characteristics of large capacity and low cost, and the battery has the characteristics of high energy density. In this paper, two kinds of energy storage devices are connected to the microgrid, and a DC microgrid model with composite energy storage is built. Four examples are designed to verify the proposed system and control strategy. The conclusions are as follows:

(1) The composite energy storage system and control strategy adopted can effectively suppress the volatility and intermittency of renewable energy and can deal with the sudden failure of the system and stabilize the DC bus voltage.

(2) The PI parameter tuning methods adopted: fuzzy PI control and PSO PI control can realize the online tuning and optimization of PI parameters and achieve better voltage stabilization effects than traditional PI control.

(3) The composite energy storage system adopted combines the advantages of AA-CAES and battery energy storage units, which can not only achieve rapid response to fluctuations, but also have sufficient capacity to support bus voltage.

(4) The composite energy storage system adopted can prolong the single charging and discharging cycle of a battery and extend its service life when the cycle times of the battery are fixed.

Future research direction: (1) The DC microgrid in this paper adopts a single-bus structure, and the optimum location and placement of the ESS has not been considered. Therefore, under the configuration of multiple DC microgrids, the impact of the optimum location and placement of ESS on the power grid operation capacity will be studied. (2) In this paper, fixed-line rating is considered. Later research can add an elastic rating system to the model to verify the reliability of the proposed control method. (3) The problem of battery degradation is not considered in this paper. The impact of ESS stability on the microgrid can be further studied in the future.

**Author Contributions:** Z.Y.: Conceptualization, Software, Formal analysis, Data Curation, Writing—Original draft preparation, Visualization. C.W.: Methodology, Software, Formal analysis, Data Curation, Writing—Original draft preparation, Visualization. J.H.: Conceptualization, Methodology, Writing—Review and Editing. F.Y.: Methodology, Writing—Review and Editing. Y.S.: Methodology, Writing—Review and Editing. H.M.: Methodology, Writing—Review and Editing. W.H.: Methodology, Writing—Review and Editing. H.S.: Conceptualization, Methodology, Writing—Review and Editing. All authors have read and agreed to the published version of the manuscript.

**Funding:** This research was funded by the State Grid Hubei Electric Power Company Science and Technology Project (No. 521532220005).

**Data Availability Statement:** Not applicable.

**Conflicts of Interest:** The authors declare no conflict of interest.

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
