# Peer review of "Analysis of Voltage Control Strategies for DC Microgrid with Multiple Types of Energy Storage Systems"

_electronics, doi:10.3390/electronics12071661_

Round 1

Reviewer 1 Report

This is a highly relevant issue at the current moment in electrical energy systems, in which the scientific world is studying how to reduce polluting fuels with the use of renewable energies. And one of the techniques in evidence is using the bakcup support with energy storage by batteries. In this paper, the performances of three voltage control strategies for DC microgrid are compared, including the proportion integration (PI) control, the fuzzy PI control and particle swarm optimization (PSO) PI control.

The authors mention in the Introduction section that "the effects of light intensity fluctuation and power generation unit failure on system stability are simulated respectively, and the effectiveness and reliability of the strategy are verified." Question: Were there measurements numerically proving the failure rate, reliability and efficiency? If not, could the authors explain how these gains were made?

The bibliographic review lacks real-time simulations with the use of battery supply using renewable sources in industrial and commercial situations or energy substations, showing the valuable techniques so well developed here. I leave as a bibliographic suggestion some papers that can contribute to enhance even more the work developed:

1) 10.3390/su142113765: Development of Operation Strategy for Battery Energy Storage System into Hybrid AC Microgrids

2) 10.3390/en16031175: Development of a Method for Sizing a Hybrid Battery Energy Storage System for Application in AC Microgrid

3) 10.3390/en15249514: Case Study of Backup Application with Energy Storage in Microgrids

4) 10.3390/en16031468: Operational Data Analysis of a Battery Energy Storage System to Support Wind Energy Generation

The conclusions with the 4 examples to verify the proposed system and the control strategy had interesting and significant results.

Congratulations to the authors!

Author Response

Please see the attached response document. Thank you very much.

Reviewer 2 Report

In recent decades, DC microgrids have garnered worldwide attention due to their high system efficiency and ease of control. Voltage control is a critical factor in ensuring the successful integration of renewable energy into DC microgrids as a self-sufficient energy system, and energy storage systems (ESSs) are often used to regulate power fluctuations and maintain voltage stability. This study compares three voltage control strategies for DC microgrids - proportion integration (PI) control, fuzzy PI control, and particle swarm optimization (PSO) PI control. The impact of two types of ESSs - batteries and advanced adiabatic compressed air energy storage (AA-CAES) - on voltage control is investigated, and the control performance is compared under various scenarios, including renewable energy fluctuations, the involvement of ESSs, and different fault conditions. The results demonstrate the effectiveness of the control strategies and the suitability of incorporating ESSs in DC microgrids.

I think the paper merits a publication in this journal, but I have the following concerns that need the authors’ response:

1)) some studies have shown that the optimum location and placement of the ESS affects its ability to participate in grid operations. May I know why has this factor been ignored in the proposed model? Although the proposed method is commendable, I think including this additional factor would have made the studies even more interesting. At least, I think the authors should discuss how this factor can be included in the proposed model, and discuss the potential impacts toward the results and discussions that have been provided so far.

2)) The flexible rating of the transmission network has been shown to improve the reliability and improve renewable integration in various studies, such as [“Reliability impacts of the dynamic thermal rating and battery energy storage systems on wind-integrated power networks”, Sustainable Energy, Grids and Networks], [“Network topology optimisation based on dynamic thermal rating and battery storage systems for improved wind penetration and reliability”, Applied Energy] and [“Impact of the real-time thermal loading on the bulk electric system reliability”, IEEE Trans Reliability]. However, the authors have considered a fixed line rating, which is less pragmatic and majorly underestimate the actual rating of lines. Hence, it is suggested that the authors additionally include the flexible rating system into their proposed model. Otherwise, they can at least discuss how this can be done and the potential impacts toward the results and discussions that have been provided.

3)) The reliability assessment of the ESS has been incorporated before in power system considering the thermally induced events, as shown in (a) [“Li, S., Ye, C., Ding, Y., & Song, Y. (2022). Reliability Assessment of Renewable Power Systems Considering Thermally-Induced Incidents of Large-Scale Battery Energy Storage. IEEE Transactions on Power Systems. doi: 10.1109/TPWRS.2022.3200952”], which was subsequently been extended to study its effects toward the stability of the power grid, as shown in (b) [“Zhong, C., Zhou, Y., Chen, J., & Liu, Z. (2022). DC-side synchronous active power control of two-stage photovoltaic generation for frequency support in Islanded microgrids. Energy Reports, 8, 8361-8371. doi: https://doi.org/10.1016/j.egyr.2022.06.030”], (c) [“Sun, B., Li, Y., Zeng, Y., Chen, J., & Shi, J. (2022). Optimization planning method of distributed generation based on steady-state security region of distribution network. Energy Reports, 8, 4209-4222. doi: https://doi.org/10.1016/j.egyr.2022.03.078”], and (d) [“Zhang, Q., Liu, Z., Jiang, X., Peng, Y., Zhu, C.,... Li, Z. (2022). Experimental investigation on performance improvement of cantilever piezoelectric energy harvesters via escapement mechanism from extremely Low-Frequency excitations. Sustainable Energy Technologies and Assessments, 53, 102591. doi: https://doi.org/10.1016/j.seta.2022.102591”]. The four studies above highlight the consideration of ESS application from the stability perspective, which has been ignored in this journal. Given the lack of such consideration, I think the authors should discuss how these factors can be included in the proposed model, and discuss the potential impacts toward the results and discussions that have been provided so far.

Author Response

(The authors gave the same response as above.)

Reviewer 3 Report

Dear authors,

For DC microgrid voltage stabilization control process, a composite energy storage system is proposed. The article is well written. The simulation results sustain the theoretical part and the comparation results between the control method.

The problems that I found are related to the written part. Please add a description or define all the abbreviation that you are using in the text, before their first use.

Please add also more recently references from 2022, 2021...

Author Response

(The authors gave the same response as above.)

Reviewer 4 Report

Figure 1. Battery arrow must be bi-directional 

section 1.2 is available in literature , provide the reference 

Figure 6. why PV current is taken as reference , voltage is not taken?

Figure . 8 the letters are hiding a bit in the box

table 3 what about the values in the second row

table 4 check it is 10 power 5 or 105, same in all the rows

Figure 42 both(a and b) may be combined and compared

Author Response

(The authors gave the same response as above.)

Round 2

Reviewer 2 Report

No more comments. Thank you

Reviewer 4 Report

English language and style are fine/minor spell check required